# COMPOSITIONAL SIMULATION-BASED INFERENCE FOR TIME SERIES

**Manuel Gloeckler** *
University of Tübingen
Tübingen, Germany
`manuel.gloeckler@uni-tuebingen.de`

**Shoji Toyota** *
Kyushu University
Fukuoka, Japan
`toyota@ait.kyushu-u.ac.jp`

**Kenji Fukumizu**
The Institute of Statistical Mathematics
Tokyo, Japan
`fukumizu@ism.ac.jp`

**Jakob H. Macke**
University of Tübingen & MPI-IS Tübingen
Tübingen, Germany
`jakob.macke@uni-tuebingen.de`

## ABSTRACT

Amortized simulation-based inference (SBI) methods train neural networks on simulated data to perform Bayesian inference. While this strategy avoids the need for tractable likelihoods, it often requires a large number of simulations and has been challenging to scale to time series data. Scientific simulators frequently emulate real-world dynamics through thousands of single-state transitions over time. We propose an SBI approach that can exploit such Markovian simulators by locally identifying parameters consistent with individual state transitions. We then compose these local results to obtain a posterior over parameters that align with the entire time series observation. We focus on applying this approach to neural posterior score estimation but also show how it can be applied, e.g., to neural likelihood (ratio) estimation. We demonstrate that our approach is more simulation-efficient than directly estimating the global posterior on several synthetic benchmark tasks and simulators used in ecology and epidemiology. Finally, we validate scalability and simulation efficiency of our approach by applying it to a high-dimensional Kolmogorov flow simulator with around one million data dimensions.

## 1 INTRODUCTION

Numerical simulations are a central approach for tackling problems in a wide range of scientific and engineering disciplines, including physics (Brehmer & Cranmer, 2022; Dax et al., 2021), molecular dynamics (Hollingsworth & Dror, 2018), neuroscience (Gonçalves et al., 2020) and climate science (Watson-Parris et al., 2021). Simulators often include at least some parameters that cannot be measured experimentally. Inferring such parameters from observed data is a fundamental challenge. Bayesian inference provides a principled approach to identifying parameters that align with empirical observations (Gelman et al., 2020). Standard algorithms for Bayesian inference, such as Markov Chain Monte Carlo (MCMC) (Gilks et al., 1995) and variational inference (Beal, 2003), generally require access to the likelihoods $p(\boldsymbol{x}|\boldsymbol{\theta})$. However, for many simulators, directly *evaluating* the likelihood remains intractable, rendering conventional Bayesian approaches inapplicable. Yet, *generating* synthetic data $\boldsymbol{x} \sim p(\boldsymbol{x}|\boldsymbol{\theta})$ is feasible for numerical simulators. Simulation-based inference (SBI) methods offer a powerful alternative to perform Bayesian inference for such simulator models with intractable likelihoods (Cranmer et al., 2020).

Classical SBI methods, like Approximate Bayesian Computation (ABC) (Beaumont et al., 2002) and synthetic likelihoods (Wood, 2010), struggle to effectively scale to high-dimensional simulations. To address this, SBI methods have been developed, which train neural networks to represent likelihoods (Papamakarios et al., 2019; Glöckler et al., 2022; Boelts et al., 2022), likelihood ratios (Durkan et al., 2020; Hermans et al., 2020; 2022; Miller et al., 2022), posteriors (Papamakarios & Murray, 2016; Lueckmann et al., 2017; Greenberg et al., 2019; Deistler et al., 2022; Geffner et al.,

---

*Equal Contributions

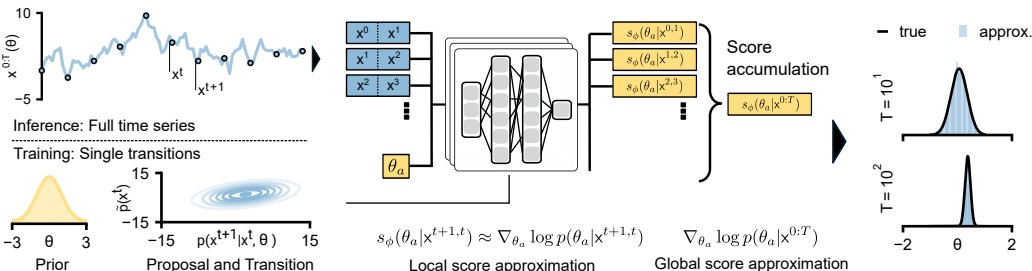

Fig 1: **Illustration of Factorized Neural Score Estimation (FNSE)**. The goal is to perform parameter inference on a full time series model. The training process uses a smaller subsets of single-state transitions initialized at arbitrary proposal $\tilde{p}(\boldsymbol{x}^t)$, with parameters sampled from a prior distribution. During inference, the time series is divided into single state transitions, and each state transition is evaluated by the neural network to estimate local posterior scores. These local estimates are then aggregated to form a global approximation, which is subsequently used to sample from the overall posterior distribution. Here, $a$ denotes the diffusion time, and $\boldsymbol{\theta}_a$ is the associated noisy parameter.

2023; Wildberger et al., 2023; Sharrock et al., 2024) or target several properties at once (Radev et al., 2023; Gloeckler et al., 2024b). These methods allow for parameter inference without requiring additional simulations after training, making them more efficient than traditional approaches (Lueckmann et al., 2021): they effectively *amortize* the cost of the simulation, and/or the full inference approach (Gershman & Goodman, 2014; Le et al., 2017).

However, applying these neural amortized methods to time series simulations can be challenging due to the high computational cost of repeated simulator calls. Running numerous time series simulations—possibly with varying sequence lengths—can be computationally prohibitive or at least wasteful. Instead, it seems advantageous to exploit the temporal structure of these simulators: Many—if not most—scientific simulators for time series data are based on (stochastic) differential equations that model dynamics of a system through state transitions over time (e.g., Sedlmeier et al. (2016); Goswami et al. (2006); Strauss & Effenberger (2017); Sirur et al. (2016)), or directly model processes which are iteratively updated at each time-step. As a consequence, such simulators have an inherently Markovian structure, which can be leveraged for efficient inference! In particular, we can approach the problem on the level of single-state transitions, a simpler task that can be tackled with fewer simulations. For instance, in the Kolmogorov Flow simulator (Sec. 4.4), the simulation outputs reach nearly one million dimensions for a time series of length hundred, which is out of range for most direct SBI approaches. In contrast, the dimensionality of single state transitions is only a few thousand.

In this work, we propose a method for efficient simulation-based inference for Markovian time series simulators. Unlike other neural SBI methods that require long sequences for training, our approach locally estimates parameters consistent with a single state transition. By aggregating these local solutions, we recover a global posterior approximation for time series of arbitrary length, significantly reducing simulation costs (Fig. 1). We focus on applying our strategy to posterior score-based approaches (FNSE, Factorized Neural Score Estimation) and additionally apply it to likelihood(-ratio) estimation methods (FNLE and FNRE). A related challenge has been addressed in prior studies (Geffner et al., 2023; Linhart et al., 2024; Boelts et al., 2022) which had focused on efficient inference of i.i.d. (independently and identically distributed) observations. We here show that these approaches can be extended to Markovian simulators—arguably the dominant model-class for mechanistic simulators for time-series—and empirically evaluate our approach on a series of synthetic benchmark tasks, as well as simulators from ecology and epidemiology. Finally, we demonstrate the scalability and efficiency of this approach on the high-dimensional Kolmogorov flow problem.

## 2 PRELIMINARIES

### 2.1 PROBLEM SETTING

We target simulators $p(\boldsymbol{x}^{0:T}|\boldsymbol{\theta})$ that output time series $\boldsymbol{x}^{0:T} := (\boldsymbol{x}^0, ..., \boldsymbol{x}^T) \in \mathbb{R}^{T \times d}$. Many simulators in scientific fields are implemented using (stochastic) differential equations, which inherently satisfy the Markov property. We focus on this class of simulators, where the likelihood takes the

following form

$$p(\boldsymbol{x}^{0:T}|\boldsymbol{\theta}) = p(\boldsymbol{x}^0|\boldsymbol{\theta}) \prod_{t=0}^{T-1} p(\boldsymbol{x}^{t+1}|\boldsymbol{x}^t, \boldsymbol{\theta}). \tag{1}$$

To simplify the exposition, we constrain the initial condition to be independent of the parameter, $p(\boldsymbol{x}^0|\boldsymbol{\theta}) = p(\boldsymbol{x}^0)$, and assume that the simulation transition $p(\boldsymbol{x}^{t+1}|\boldsymbol{x}^t, \boldsymbol{\theta})$ does not depend on time $t$. We relax both of these assumptions in Appendix A.2, and also address the general case of higher-order Markov chains in Appendix A.2.3. We assume one can sample from the state-transition function $\boldsymbol{x}^{t+1} \sim \mathcal{T}(\boldsymbol{x}^{t+1}|\boldsymbol{x}^t, \boldsymbol{\theta})$, but not evaluate its probability density.

## 2.2 Amortized methods for simulation-based inference

**Neural Posterior Estimation (NPE):** Neural Posterior Estimation (NPE; Papamakarios & Murray (2016); Lueckmann et al. (2017); Greenberg et al. (2019); Wildberger et al. (2023)) methods typically use a conditional neural density estimator, e.g., a normalizing flow (Rezende & Mohamed, 2015; Papamakarios et al., 2017; Durkan et al., 2019) to approximate the target posterior distribution. The model $q_\phi(\boldsymbol{\theta}|\boldsymbol{x}^{0:T})$ is trained via maximum likelihood to estimate $p(\boldsymbol{\theta}|\boldsymbol{x}^{0:T})$ from a dataset of parameter-data pairs $\{(\boldsymbol{\theta}_i, \boldsymbol{x}_i^{0:T})\}_{i=1}$, where $(\boldsymbol{\theta}_i, \boldsymbol{x}_i^{0:T}) \sim p(\boldsymbol{\theta})p(\boldsymbol{x}^{0:T}|\boldsymbol{\theta})$. For observations of inhomogeneous size, NPE often incorporates an embedding network. Embedding networks can be designed as recurrent networks (Lueckmann et al., 2017), permutation-invariant architectures (Chan et al., 2018; Radev et al., 2020), or attention-based models (Schmitt et al., 2023) to respect known invariances in the data. At inference time, NPE yields an amortized approximation of the posterior for any observation $\boldsymbol{x}_o^{0:T}$ therefore offering rapid evaluations or samples of the posterior.

**Neural likelihood(-ratio) estimation (NLE, NRE):** Neural Likelihood(-Ratio) Estimation (NLE (Papamakarios et al., 2019), NRE (Hermans et al., 2020; 2022; Miller et al., 2022)) methods do not directly approximate the posterior distribution but instead train a surrogate $q_\psi(\boldsymbol{x}_i^{0:T}|\boldsymbol{\theta})$ for the likelihood $p(\boldsymbol{x}_i^{0:T}|\boldsymbol{\theta})$ or likelihood ratio $p(\boldsymbol{x}_i^{0:T}|\boldsymbol{\theta})/p(\boldsymbol{x}_i^{0:T})$. Once the likelihood is accessible via the surrogate, standard techniques such as MCMC or variational inference can be applied to perform inference. These approaches circumvent the problem of not having access to the likelihood, but still leaves us with the drawbacks of standard inference methods, i.e., slow sampling and potential failure modes such as robustly handling multimodality (Glöckler et al., 2022). Nonetheless, they also inherit the flexibility of standard inference techniques, such as the ability to handle multiple independent (or conditionally independent) observations by simply multiplying the likelihood terms (Boelts et al., 2022).

**Neural score estimation (NSE):** Score-based diffusion models are a powerful tool for generating samples from a target distribution. Common diffusion models (Song & Ermon, 2019; Song et al., 2021; Ho et al., 2020) are based on stochastic differential equations (SDEs) that can be expressed as $d\boldsymbol{\theta}_a = f(a)\boldsymbol{\theta}_a da + g(a)d\boldsymbol{w}$[1] with $\boldsymbol{w}$ being a standard Wiener process. The drift and diffusion coefficients $f$ and $g$ are chosen such that the solution to this SDE defines a diffusion process that transforms any initial distribution into a simple noise distribution $p_A(\boldsymbol{\theta}_A) = \mathcal{N}(\boldsymbol{\theta}_A, \mu_A, \sigma_A^2 I)$. Samples from any target, such as the posterior $p(\boldsymbol{\theta}|\boldsymbol{x}^{0:T})$, can then be obtained by simulating the reverse diffusion process (Anderson, 1982)

$$d\boldsymbol{\theta}_a = \left\{ f(a) - g^2(a) \cdot \nabla_{\boldsymbol{\theta}_a} \log p(\boldsymbol{\theta}_a|\boldsymbol{x}^{0:T}) \right\} da + g(a)d\tilde{\boldsymbol{w}}, \tag{2}$$

where $\tilde{\boldsymbol{w}}$ is a backward-in-time Wiener process. In practice, we can not access the analytic form of the score $s(\boldsymbol{\theta}_a|\boldsymbol{x}^{0:T}) = \nabla_{\boldsymbol{\theta}_a} \log p(\boldsymbol{\theta}_a|\boldsymbol{x}^{0:T})$, but we can estimate it from samples using conditional denoising score-matching (Hyvärinen & Dayan, 2005; Song et al., 2021)

$$\mathcal{L}(\phi) = \mathbb{E}_{a,\boldsymbol{\theta},\boldsymbol{\theta}_a,\boldsymbol{x}^{0:T}} \left[ \lambda(a) || s_\phi(\boldsymbol{\theta}_a|\boldsymbol{x}^{0:T}) - \nabla_{\boldsymbol{\theta}_a} \log p(\boldsymbol{\theta}_a|\boldsymbol{\theta}) ||_2^2 \right]$$

where $\lambda$ denotes a positive weighting function and $p(\boldsymbol{\theta}_a|\boldsymbol{\theta}) = \mathcal{N}(\boldsymbol{\theta}_a; s(a)\boldsymbol{\theta}, \sigma(a)^2 \boldsymbol{I})$.

This recently proposed approach has been highly successful across various tasks (Geffner et al., 2023; Sharrock et al., 2024; Gloeckler et al., 2024b). It offers a trade-off between the efficiency of the static NPE method and the flexibility of slower but more flexible NL(R)E method. By enabling feasible post-hoc modifications and composability through appropriate adjustments to the backward diffusion process, it bridges the gap between these two approaches (Geffner et al., 2023; Gloeckler et al., 2024b).

---

[1]The diffusion time is denoted by $a$, against convention, to distinguish it from $t$ in the time series.

---

**Algorithm 1** Training

1: **Input:** prior $p(\boldsymbol{\theta})$, proposal $\tilde{p}(\boldsymbol{x}^t)$, transition function $\mathcal{T}(\boldsymbol{x}^{t+1}|\boldsymbol{x}^t, \boldsymbol{\theta})$, score net $s_\phi$.
2: $\mathcal{D} = \emptyset$    // Generate training dataset
3: **for** $i = 1$ to $N$ **do**
4:     $\boldsymbol{\theta}_i \sim p(\boldsymbol{\theta}); \quad \boldsymbol{x}_i^t \sim \tilde{p}(\boldsymbol{x}^t)$
5:     $\boldsymbol{x}_i^{t+1} \sim \mathcal{T}(\boldsymbol{x}^{t+1}|\boldsymbol{x}_i^t, \boldsymbol{\theta}_i)$
6:     $\mathcal{D} = \mathcal{D} \cup \{(\boldsymbol{\theta}_i, \boldsymbol{x}_i^t, \boldsymbol{x}_i^{t+1})\}$
7: **end for**
8: Train $s_\phi$ by minimizing Eq. 4 using $\mathcal{D}$

---

**Algorithm 2** Evaluation

1: **Input:** prior $p(\boldsymbol{\theta})$, observation $\boldsymbol{x}_o^{0:T}$, `compose` method (see Sec. 3.2.2).
2:
3: **def** $s_{\text{glob}}(\boldsymbol{\theta}_a, \boldsymbol{x}_o^{0:T})$:
4:     $s_{\text{local}} = []$
5:     **for** $t = 0$ to $T$ **do**
6:         $s_{\text{local}} \mathrel{+}= [s_\phi(\boldsymbol{\theta}_a, \boldsymbol{x}_o^{t,t+1})]$
7:     $s_{\text{glob}} = $`compose`$(s_{\text{local}}, p(\boldsymbol{\theta}))$
8:     **return** $s_{\text{glob}}$
9: Sample $p(\boldsymbol{\theta}|\boldsymbol{x}_o^{0:T})$ via $s_{\text{glob}}$

---

# 3 METHODS

## 3.1 GLOBAL INFERENCE FROM SINGLE-STEP TRANSITIONS

Direct estimation of the *global* target distribution (i.e., the posterior $p(\boldsymbol{\theta}|\boldsymbol{x}^{0:T})$ in NPE or the likelihood $p(\boldsymbol{x}^{0:T}|\boldsymbol{\theta})$ in NLE; Sec. 2.2) often requires a large number of simulations of the entire time series, leading to intractable, or at least very expensive, estimation problems. We mitigate the problem of expensive simulation calls by leveraging the Markov factorization of the forward model: Since the simulator is completely specified by its state-transition probabilities, $p(\boldsymbol{x}^{t+1}|\boldsymbol{x}^t, \boldsymbol{\theta})$, these also contain all relevant information about the parameters. It is therefore possible to recover the global target distribution (e.g., the posterior) for a time series of variable size given a dataset of single-step transitions. Building on this observation, we aim to estimate *local* target scores $s_{\text{local}}$ using single-step transition simulation data $\mathcal{D} = \{(\theta_i, \mathbf{x}_i^{t:t+1})\}_{i=1}^N$ (Alg. 1, Sec. 3.2.1). Afterwards, we recover the global target distribution by aggregating these local solutions using a composition rule (`compose`, Alg. 2, Sec. 3.2.2). In the following, we apply this approach to NSE, NLE, and NRE (Sec. 3.2, 3.3) and discuss the design of the required proposal $\tilde{p}(\boldsymbol{x}^t)$ (Sec. 3.4) for sampling single-state transitions.

## 3.2 FACTORIZED NEURAL SCORE ESTIMATION (FNSE)

### 3.2.1 LOCAL SCORE ESTIMATION

Assuming that $p(\boldsymbol{x}^{0:T}|\boldsymbol{\theta})$ satisfies Eq. 1 and $p(\boldsymbol{x}^0|\boldsymbol{\theta}) = p(\boldsymbol{x}^0)$, the global target $\nabla_{\boldsymbol{\theta}} \log p(\boldsymbol{\theta}|\boldsymbol{x}^{0:T})$ in NSE can be factorized as

$$\nabla_{\boldsymbol{\theta}} \log p(\boldsymbol{\theta}|\boldsymbol{x}_o^{0:T}) = \sum_{t=0}^{T-1} \nabla_{\boldsymbol{\theta}} \log \tilde{p}(\boldsymbol{\theta}|\boldsymbol{x}_o^t, \boldsymbol{x}_o^{t+1}) - (T-1) \cdot \nabla_{\boldsymbol{\theta}} \log p(\boldsymbol{\theta}). \tag{3}$$

Here, $\tilde{p}(\boldsymbol{\theta}|\boldsymbol{x}^t, \boldsymbol{x}^{t+1})$ is a posterior with $\boldsymbol{x}^t$ following any proposal distribution $\boldsymbol{x}^t \sim \tilde{p}(\boldsymbol{x}^t)$ and $\boldsymbol{x}^{t+1}$ following the state transition $\boldsymbol{x}^{t+1} \sim \mathcal{T}(\boldsymbol{x}^{t+1}|\boldsymbol{x}^t, \boldsymbol{\theta})$ (Appendix Sec. A.1). The factorization (3) implies that the global posterior is fully characterized by $s(\boldsymbol{\theta}|\boldsymbol{x}^t, \boldsymbol{x}^{t+1}) = \nabla_{\boldsymbol{\theta}} \log \tilde{p}(\boldsymbol{\theta}|\boldsymbol{x}^t, \boldsymbol{x}^{t+1})$. We can estimate this quantity using only single-state transitions by minimizing the loss

$$\mathcal{L}(\phi) = \mathbb{E}_{a, \boldsymbol{\theta}, \boldsymbol{\theta}_a, \boldsymbol{x}^t, \boldsymbol{x}^{t+1}} \left[ \lambda(a) || s_\phi(\boldsymbol{\theta}_a | \boldsymbol{x}^t, \boldsymbol{x}^{t+1}) - \nabla_{\boldsymbol{\theta}_a} \log p(\boldsymbol{\theta}_a | \boldsymbol{\theta}) ||_2^2 \right], \tag{4}$$

given a proposal distribution $\boldsymbol{x}^t \sim \tilde{p}(\boldsymbol{x}^t)$. As a result, we can learn a *local* score estimator by empirically minimizing this loss given a dataset of single-step simulations (Alg. 1). For a globally amortized posterior approximation, the proposal distribution must at least satisfy two properties: *(i)* it must have support at any point $\boldsymbol{x}^t$ the simulator can attain, *(ii)* the proposed point $\boldsymbol{x}^t$ must be independent of the parameter $\boldsymbol{\theta}$ involved in the transition to $\boldsymbol{x}^{t+1}$ to ensure Eq. 3 remains valid (Appendix Sec. A.1). These constraints leave quite some flexibility in the choice of the proposal. Its choice will have a significant impact, given a finite simulation budget (Sec. 3.4). Our approach can readily be extended to higher-order Markov chains, which require proposing multiple past observations (Appendix Sec. A.2.3). Between the two extreme cases of learning from single-state transitions, or full time series, there is a continuum of approaches, similarly to what was shown in Geffner et al. (2023) for independent observations. We can extend our results to amortizing over any finite number of transitions, leading to a partially factorized approach (Appendix Sec. A.2.4).

### 3.2.2 LOCAL SCORE COMPOSITION

At inference time, we need to recover the global score $s_{\text{glob}}(\boldsymbol{\theta}_a|\boldsymbol{x}_o^{0:T}) = \nabla_{\boldsymbol{\theta}_a} \log p(\boldsymbol{\theta}_a|\boldsymbol{x}_o^{0:T})$, by composing all local scores $s_{\text{local}}(\boldsymbol{\theta}_a|\boldsymbol{x}_o^{t:t+1}) = \nabla_{\boldsymbol{\theta}_a} \log p(\boldsymbol{\theta}_a|\boldsymbol{x}_o^{t:t+1})$ estimated by the network, thereby defining the `compose` function in Alg. 2. Finally, the composed global score $s_{\text{glob}}$ will be passed to a diffusion sampler to obtain samples from the target posterior $p(\boldsymbol{\theta}|\boldsymbol{x}_o^{0:T})$.

While the composition as introduced in Eq. 3 is valid for $a = 0$, it becomes invalid for any $a > 0$ (i.e., each noisy posterior). The reason for this is the following: The likelihood for the 'noisy' parameter $p(\boldsymbol{x}^{0:T}|\boldsymbol{\theta}_a)$ no longer satisfies the Markov property, even if $p(\boldsymbol{x}^{0:T}|\boldsymbol{\theta})$ does (Weilbach et al., 2023; Gloeckler et al., 2024b; Geffner et al., 2023; Rozet & Louppe, 2023). In the i.i.d. setting, this issue has been tackled by Linhart et al. (2024), and we will adapt their approach to the Markov setting. The correct global score is intractable in most cases, but they proposed an approximation that can be directly adapted for Markovian time series:

$$\nabla_{\boldsymbol{\theta}_a} \log p(\boldsymbol{\theta}_a|\boldsymbol{x}^{0:T}) \approx \boldsymbol{\Lambda}(\boldsymbol{\theta}_a)^{-1} \left( \sum_{t=0}^{T-1} \boldsymbol{\Sigma}_{a,t,t+1}^{-1} s_\phi(\boldsymbol{\theta}_a|\boldsymbol{x}^t, \boldsymbol{x}^{t+1}) + (1-T)\boldsymbol{\Sigma}_a^{-1} \nabla_{\boldsymbol{\theta}_a} \log p(\boldsymbol{\theta}_a) \right),$$

where $\boldsymbol{\Lambda}(\boldsymbol{\theta}_a) = \sum_{t=0}^{T-1} \boldsymbol{\Sigma}_{a,t,t+1}^{-1} + (1-T)\boldsymbol{\Sigma}_a^{-1}$. Here $\boldsymbol{\Sigma}_a$ denotes the covariance matrix of $p(\boldsymbol{\theta}|\boldsymbol{\theta}_a)$ (the usually tractable denoising prior) and $\boldsymbol{\Sigma}_{a,t,t+1}$ denotes the covariance matrix of $p(\boldsymbol{\theta}|\boldsymbol{\theta}_a, \boldsymbol{x}^{t,t+1})$ (the denoising posterior), which we need to estimate. This can be done by estimating the posterior covariance from samples of the local posteriors obtained via the diffusion model, referred to as **GAUSS**. Unless otherwise specified, we use this approximation as the default composition rule for FNSE. Alternatively, we can estimate it via Tweedie's moment projection using the Jacobian of the score estimator, referred to as **JAC** (Linhart et al. (2024), see Appendix B for details). In contrast, Geffner et al. (2023) addressed this issue (in the i.i.d. setting) through post-hoc sampling corrections (we referred to this uncorrected variant as **FNPE**, Appendix Sec. B.2 for details)

### 3.3 FACTORIZED LIKELIHOOD(-RATIO) ESTIMATION (FNLE, FNRE)

For likelihood(-ratio) estimation, applying our approach is straightforward: by assumption, the likelihood factorizes as shown in Eq. 1. As a consequence, we can directly learn the transition density $p(\boldsymbol{x}^{t+1}|\boldsymbol{x}^t, \boldsymbol{\theta})$, similarly to the method described in Alg. 1, by replacing the score matching loss with the appropriate likelihood (or likelihood-ratio) loss (Papamakarios et al., 2019; Hermans et al., 2020; 2022; Miller et al., 2022). Once the transition density is obtained, the global log-likelihood approximation $\ell_{\text{glob}}$ can be computed directly from Eq. 1 (i.e., by simply summing up the local approximations). Classical MCMC techniques can be employed for sampling. We use reference implementations of NLE and NRE as implemented in the *sbi* package (Tejero-Cantero et al., 2020b; Boelts et al., 2024), adapted to the Markovian setting.

### 3.4 CONSTRUCTION OF THE PROPOSAL DISTRIBUTION

A proposal $\tilde{p}(\boldsymbol{x}^t)$ for single state transitions needs to satisfy conditions *(i)* and *(ii)* in Sec. 3.2.1, but solely relying on these properties does not necessarily make up a "good" proposal. Essentially, it specifies which regions of the data domain are represented in the training dataset. Therefore, a better local posterior approximation is expected for states that are more likely to be generated by the proposal compared to those that are less likely. This provides an opportunity and challenge to design appropriate proposals for a given simulator.

For an amortized posterior approximation, the proposal distribution design should be guided by the prior predictive distribution. The resulting trajectories $\boldsymbol{x}^{0:T} \sim p(\boldsymbol{x}^{0:T})$ encompass likely states $\boldsymbol{x}^t$ and we can use this to construct a reasonable proposal $\tilde{p}(\boldsymbol{x}^t)$ (e.g. by simply proposing these at random)—however, this approach comes at the cost of upfront simulations. Alternatively, we can construct the proposal $\tilde{p}(\boldsymbol{x}^t)$ based on observations available prior to training. This will focus simulation resources on the intended application but might not yield a good amortized approximation. For further details on proposal design, see Appendix Sec. C.

## 4 EMPIRICAL RESULTS

### 4.1 EVALUATION APPROACH

We empirically assess the accuracy and computational efficiency of the proposed approach, comparing it against non-factorized NPE with an appropriately chosen embedding net as a baseline. We note that the simulation budget is not determined by the number of simulator calls $N$, but rather by the number of calls to the state-transition function, i.e., $T \cdot N$. We, therefore, configure the NPE baseline with a $T_{\max} = 10$ steps, i.e., if the simulation budget is $10k$, the NPE baseline used 1000 simulations, each of which is simulated for 10 steps (experiments with longer segments in Appendix E). We additionally use data augmentation by duplicating shortened variants ($T < T_{\max}$) of these simulations. This aims to help the RNN embedding net to generalize to different sequence lengths. To also amortize over different initial conditions, the initial condition for each ten-step simulation was drawn from the proposal also used in the factorized methods. As main metrics for comparison, we use sliced Wasserstein distance ($sW_1$) and Classifier two sample test accuracy (C2ST) on reference posterior samples (Lopez-Paz & Oquab, 2016; Kolouri et al., 2019; Bischoff et al., 2024). We average the estimated value over a total of 10 randomly drawn observations. The whole process, i.e., training, sampling, and evaluation, was repeated five times, starting from different random seeds.

We begin by evaluating the methods on a set of newly designed benchmark tasks for Markovian simulators (Sec. 4.2). Next, we apply the approach to classical models from ecology and epidemiology, including the stochastic Lotka-Volterra and SIR models (Sec. 4.3). Finally, we demonstrate the scalability of the method on a large-scale Kolmogorov flow task, where we perform inference on very high-dimensional data using only 200k simulator steps (Sec. 4.4).

### 4.2 BENCHMARKS

To investigate several properties of the proposed approach, we develop several synthetic tasks of first-order Markovian time series with associated reference posterior samplers (details in Appendix D.1).

**Gaussian RW:** A Gaussian Random Walk of form $\boldsymbol{x}^{t+1} = \alpha \cdot \boldsymbol{x}^t + \boldsymbol{\theta} + \epsilon$ for $\epsilon \sim \mathcal{N}(0, \boldsymbol{I}), \alpha < 1$ with $\boldsymbol{x}^t \in \mathbb{R}^d$ and $\boldsymbol{\theta} \in \mathbb{R}^d$. This simple task offers an analytic Gaussian posterior.

**Mixture RW:** A Mixture of Gaussian Random Walk of form $\boldsymbol{x}^{t+1} = \boldsymbol{x}^t + u \cdot \boldsymbol{\theta} + \epsilon$ for $\epsilon \sim \mathcal{N}(0, \boldsymbol{I}), u \sim \mathrm{Unif}(\{-1., 1.\})$ and $\boldsymbol{x}^t, \boldsymbol{\theta} \in \mathbb{R}^d$. By design, this task has a mixture of Gaussian transition density and a non-Gaussian bimodal posterior.

**Periodic/Linear SDE:** Both tasks are governed by the linear SDE $d\boldsymbol{x}^t = \boldsymbol{A}(\boldsymbol{\theta})\boldsymbol{x}^t + \boldsymbol{B}(\boldsymbol{\theta})d\boldsymbol{w}^t$. The periodic SDE is oscillatory with $\boldsymbol{\theta}, \boldsymbol{x}^t \in \mathbb{R}^2$ and a posterior with four modes. The linear SDE involves $\boldsymbol{A}$ and $\boldsymbol{B}$ with $\boldsymbol{x}^t \in \mathbb{R}^3$ and $\boldsymbol{\theta} \in \mathbb{R}^{18}$.

**Double well:** A nonlinear SDE $d\boldsymbol{x}^t = \theta_1 \boldsymbol{x}^t + \theta_2 \left(\boldsymbol{x}^t\right)^3 + \sigma d\boldsymbol{w}^t$, which samples from a double-well potential with modes position depending on $\theta_1, \theta_2$ (Singer, 2002; Cai et al., 2023).

We first examine the scalability of all evaluated methods as a function of the length and dimensionality of the time series while maintaining a fixed simulation budget. Specifically, we assess each method on the Gaussian random walk (RW) example in 1, 2, and 10 dimensions, using a total of 10k simulations (Fig. 2a). Both FNLE and FNSE show better simulation efficiency than the NPE baselines. FNRE is more sensitive to local errors, leading to a decline in performance over a long time series. A similar observation was made by Geffner et al. (2023) for i.i.d. data. Notably, this issue is reduced in the FNLE approach, which was not analyzed in their work but successfully applied by Boelts et al. (2022). On the other hand, the computational cost escalates significantly on long time series, as evaluating the global likelihood requires $T$ forward passes for each iteration of MCMC sampling, making FNLE and FNRE relatively slow (especially NLE, as a forward pass through the normalizing flow in FNLE is more costly than the classifier used in FNRE, Fig. 2a). In contrast, the sampling cost for FSNE remains moderate due to its more efficient sampling method, while performance remains similar to or better than FNLE.

In contrast to FNLE and FNRE, the approximative score composition (for $a > 0$) might impact the sampling quality for FNSE. Therefore, we investigate the scaling behavior of different score composition techniques for long time series (Fig. 2**b**, Appendix Fig. 8**b**). Previous work within

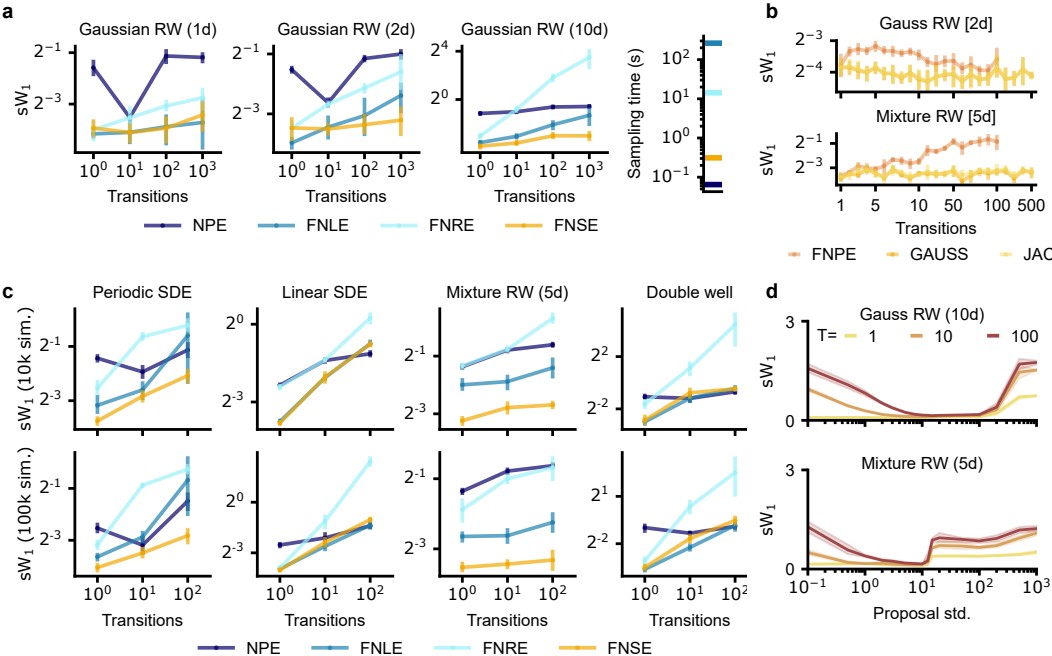

Fig 2: **Benchmarks:** We validate our method on a Gaussian Random Walk with different dimensions for different lengths (i.e. Transitions), also tracking sampling times (**a**). We assess FNSE score accumulation over Gaussian RW and Periodic SDE tasks using a fixed Euler–Maruyama sampler (**b**). We compare methods across tasks and transition steps (**c**). Finally, we examine the effect of the proposal on NFSE trained with 10k simulations from a normal proposal of varying standard deviation (**d**).

the i.i.d. case did not investigate performance beyond thirty samples (Geffner et al., 2023; Linhart et al., 2024). Specifically, for time series exceeding a length of $T > 100$, the FNPE method starts to become numerically unstable due to diverging trajectories. Although the assumptions of the GAUSS/JAC approximations are violated (as the Mixture RW has a non-Gaussian posterior), these methods still tend to perform well in practice in our implementation. Even when Langevin corrections (Song et al., 2021; Geffner et al., 2023) are introduced, the GAUSS and JAC method remain superior, especially for large $T$ (Fig. A10, Appendix Sec. B.2).

Next, we conducted benchmarks across several tasks. Overall, FNSE and FNLE show higher accuracy compared to FNRE and NPE in most tasks (Fig. 2**c**, Appendix Fig. 8**c**). An exception is the double-well task, where NPE performs relatively well due to the SDE's rapid convergence to a stationary distribution, enabling the RNN to learn a generalizable summary statistic for inference. FNSE strongly outperforms other methods in cases of multimodal transition densities (e.g., Mixture RW) and performs comparably to likelihood-based approaches in simpler cases (e.g., Periodic/Linear SDE) even if the posterior is strongly multimodal (Periodic SDE). Learning from single-step transitions is the extreme end of a spectrum of possibilities. We, therefore, also evaluate this benchmark on *partially* factorized variants using five transitions each (Appendix Fig. 9). Overall, this could improve (e.g., Double Well) or hurt (e.g., Mixture RW) performance relative to the NPE baseline. We also compared method to a broader spectrum of applicable baselines (Chen et al. (2021; 2023); Sharrock et al. (2024), Appendix Table A3, A4), and plot our results also against training budget (Appendix Fig. A6).

Finally, we investigate the impact of the proposal distribution, specifically for the FNSE method trained over 10k simulation steps (Fig. 2**d**, extended results in Fig. A11). We use a Normal distribution centered at zero, adjusting its standard deviation ranging from $\sigma = 0.1$ (too narrow) to $\sigma = 1000$ (too wide). Our results indicate a relatively broad range of standard deviations of the proposal that result in good performance. If the distribution is too narrow, the observation may reach values outside the training domain, while if it is too wide, the model is trained on values not observed during evaluation. In addition, we compared our default choice to proposals constructed using prior predictive samples (using a subset of the simulation budget, Sec. 3.4, Appendix Sec. E.4). Overall,

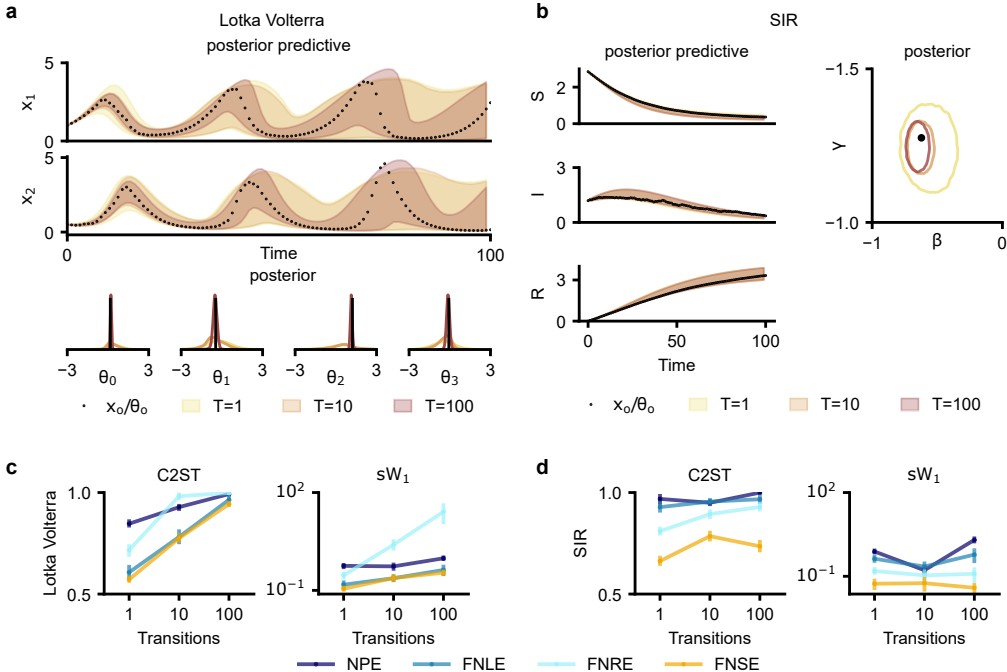

Fig 3: **Lotka Volterra and SIR experiments :** The FNSE approximate posterior (predictive) of the best performing FNSE model using 100k transition steps to train, visualized on subsequences from a fixed observation and associated true parameter. (**a**, **b**). We then show the quantitative performance in terms of C2ST and sW$_1$ for each task on ten randomly selected observations; each run is repeated five times (**c**, **d**).

this leads to similar results even if the simulation budget used to construct the proposal is subtracted from the training budget (Tab. 1)

## 4.3 LOTKA VOLTERRA AND SIR

To further evaluate our method, we tested it on two famous models from ecology and epidemiology: the Lotka-Volterra and Susceptible-Infected-Recovered (SIR) simulators. The Lotka-Volterra simulator models predator-prey dynamics through four parameters that govern prey and predator growth, hunting rates, and mortality. The SIR simulator is a fundamental model for understanding infectious disease spread. Although commonly used for benchmarking SBI on fixed-size observations Lueckmann et al. (2021), we adapted these models for our analysis (details in Appendix Sec. D.1).

We begin by demonstrating how the posteriors evolve as more time points are observed using the NFSE method (Fig. 3ab). In both tasks, we find that the posteriors converge to the true parameter values, and the posterior predictive simulations increasingly align with the time series observation. Notably, unlike in previous i.i.d. scenarios (Geffner et al., 2023; Linhart et al., 2024), the posterior uncertainty about the parameters does not necessarily decrease as additional time points are included. This is particularly evident in the SIR task, where the posterior remains largely unchanged between $T = 10$ and $T = 100$ (Fig. 3b). This can be explained by the fact that the initial dynamics contain significant information about the parameters, while the later dynamics do not, i.e., the infected population consistently declines to zero, and the susceptible and recovered populations converge to steady-state values that are independent of the parameterization.

We trained all methods on this task using $100k$ transition simulations. We then evaluate the accuracy of all models for performing inference on sequences of length $T = 1, 10, 100$. For both tasks, we see that the factorized methods (except FNRE in Lotka Volterra) generally outperform the NPE baseline (Fig. 3cd, also Appendix Fig. 6 per training budget, Appendix Tab. 3 for additional baselines). NFSE, in particular, significantly outperforms all other approaches on the SIR task. We additionally performed a simulation-based calibration analysis (Talts et al., 2018) for FNSE and NPE, which, in line with these results, showed that FNSE is better calibrated than NPE (Fig. A7.). Notably, on the SIR task, we found NPE—in contrast to FNSE—to be unable to generalize to longer time

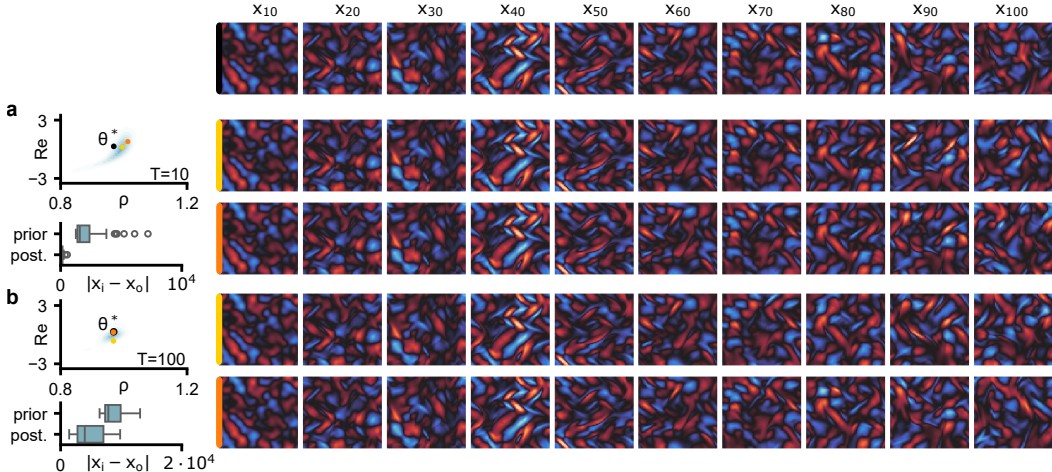

Fig 4: **Kolmogorov flow experiment:** Single example observation of length hundred (top row). We visualize the posterior distribution condition on the observation up to $t = 10$ (**a**, left top), along with a quantitative comparison of the mean absolute error (MAE) between posterior predictive samples and prior predictive samples on fifty different observations (left, bottom), including the one shown above. The vorticity of two selected predictive samples is visualized on the right. This analysis is then repeated for the entire observation (**b**).

series. In addition, we compared our default proposal to one constructed using simulations. For the Lotka Volterra task, this constructed proposal performed significantly better than our simple default (Appendix Tab. 2, Fig. 5).

## 4.4 KOLMOGOROV FLOW

Finally, we consider a task with very high dimensional observations, previously considered in the context of data-assimilation (Rozet & Louppe, 2023). The simulator models incompressible fluid dynamics governed by the Navier-Stokes equations:

$$\frac{\partial \mathbf{u}}{\partial t} = -\mathbf{u} \cdot \nabla \mathbf{u} + \frac{1}{\mathrm{Re}} \nabla^2 \mathbf{u} - \frac{1}{\rho} \nabla p + \mathbf{f} \qquad \nabla \cdot \mathbf{u} = 0,$$

where $\mathbf{u}$ is the velocity field, Re is the Reynolds number, $\rho$ is the fluid density, $p$ is the pressure field, and $\mathbf{f}$ is the external forcing. We added Gaussian noise with standard deviation $\sigma = 5 \cdot 10^{-3}$ after each transition. Following Kochkov et al. (2021); Rozet & Louppe (2023), we consider a two-dimensional domain $[0, 2\pi]^2$ with periodic boundary conditions and an external forcing $\mathbf{f}$ that corresponds to Kolmogorov forcing with linear damping Chandler & Kerswell (2013); Boffetta & Ecke (2012). We take the Reynolds number and density as free parameters $\boldsymbol{\theta} = (\mathrm{Re}, \rho)$.

In contrast to Rozet & Louppe (2023), we consider the problem of parameter inference on $\boldsymbol{\theta}$. We apply FNSE to this problem, using only *200k transition evaluations*, which successfully recovers an amortized posterior estimator that generalizes to long-term observations (Fig. 4). Notably, in such a high-dimensional state space, a simple proposal distribution would be ineffective. But, we can use both the initial distribution and simulator to propose a variety of feasible states (details in Appendix Sec. E.5). The posterior distributions for observations with both $T = 10$ and $T = 100$ are well-concentrated around the true parameter values that generated the specific observations (Fig. 4a,b). Posterior predictive samples are, on average, significantly closer to the observed data compared to prior predictives (in mean absolute error calculated over 1000 predictive samples from 50 different observations). Although the difference is less pronounced for $T = 100$, this is primarily due to the divergence for slight parameter modification on long simulations. To further investigate the performance, we perform a simulation-based calibration analysis (Talts et al., 2020) (Fig. 12). Overall, given the constraints on the simulation budget, the posterior calibration is okay but deteriorates for larger time series, indicating more simulations might be necessary to improve performance on $T = 100$.

## 5 DISCUSSION

### 5.1 RELATED WORK

We build upon previous work in the i.i.d. observation setting (Geffner et al., 2023; Linhart et al., 2024; Boelts et al., 2022), extending these methods to Markovian time series. In the context of data assimilation, Rozet & Louppe (2023) introduced a method to learn a *local* score network to estimate $p(x^{0:T})$ using a neighborhood $x^{t-k:t+k}$. In contrast, we focus on solving a *local* inverse problem that leads to a global solution. Other work aims to incorporate the task structure (approximately) into the estimating neural network architecture (Weilbach et al., 2020; 2023; Gloeckler et al., 2024a). A different line of work aims to tackle high-dimensional SBI using summary statistics. These can be handcrafted (Alsing & Wandelt, 2018), learned (Chen et al., 2021; 2023) or based on path signatures (Dyer et al., 2021a; 2022). However, estimating statistics that generalize across sequence lengths from sparse training can be challenging. We enhance the simulation efficiency of amortized SBI by leveraging the Markovian structure, which is different from sequential training schemes (Glöckler et al., 2022; Sharrock et al., 2024; Greenberg et al., 2019; Durkan et al., 2020; Deistler et al., 2022; Gutmann et al., 2016; Warne et al., 2022), which adaptively make simulations more informative for specific observations (but are thus not amortized). Similarly, other lines of work aim to distribute simulations based on a cost function (Bharti et al., 2024) and increase simulation diversity (Niu et al., 2023). The simulation efficiency of our approach could be further enhanced by incorporating such techniques. Recently, a simulation-based conditional kernel density approximation for inference in SDEs was proposed (Cai et al., 2023), which can be viewed as a special case of FNLE, substituting normalizing flows with kernel density estimation. Efficient inference in time series models was also previously approached in the ABC setting (Warne et al., 2022; Prangle, 2016; Bernton et al., 2019; Dyer et al., 2021b).

Moreover, any Kalman filtering method (Kalman, 1960; Wan & Van Der Merwe, 2000; Arasaratnam & Haykin, 2009; Julier & Uhlmann, 1997) is inherently related to FNLE, as these methods can approximate the marginal log-likelihood in an online (non-amortized) manner, but do require the availability of likelihoods (and are used to perform inference e.g. (Brouste et al., 2014)). The field of amortized inference with tractable likelihoods is inherently related (Le et al., 2017; Choi et al., 2019; Ganguly et al., 2023; Margossian & Blei, 2024).

### 5.2 LIMITATIONS AND FUTURE WORK

Our primary focus here was on inference for Markovian stochastic processes that share the same transition distribution across time. We also sketched extensions to cases where it varies with time, as well as to processes with parameterized initial distributions. We note that Hidden Markov Models, although widely applied in practical scenarios, do not fall within the scope of our method. In combination with data assimilation techniques (Rozet & Louppe, 2023) the Markovian hidden state can be recovered to which our technique can be directly applied. Extending our approach to more general probabilistic models remains a direction for future research.

The score corrections caused by score aggregation described in Subsection 3.2.2 could be further improved. In their methods, Linhart et al. (2024) approximates the data distribution by a single Gaussian distribution, which can be violated in real-world scenarios. Our results demonstrate that this approximation empirically still performs well even if these assumptions are violated.

### 5.3 CONCLUSIONS

We introduced a simulation-efficient approach for amortized inference in Markovian simulators. Although flexible embedding networks are commonly used for high-dimensional time series (Lueckmann et al., 2017; Radev et al., 2020; Schmitt et al., 2023), they often demand extensive amounts of simulations and can be fragile when faced with data perturbations (Cannon et al., 2022; Gloeckler et al., 2023). Success in these methods hinges on the embedding network's ability to capture robust and generalizable representations. In contrast, our approach decomposes the inference task into smaller, locally solvable problems, reducing computational costs and enhancing scalability for large-scale, complex simulations.

SOFTWARE AND DATA

We use `JAX` (Bradbury et al., 2018) as the computational backbone and `hydra` (Yadan, 2019) to track configurations. For reference implementation of baselines, we use `sbi` (Tejero-Cantero et al., 2020a; Boelts et al., 2024). Code to reproduce the experiments can be found in `https://github.com/mackelab/markovsbi/`.

AUTHOR CONTRIBUTIONS

Conceptualization: MG, ST, KF, JHM. Methodology: MG, ST. Software: MG. Investigation and Analysis (Benchmarks): MG, ST. Investigation and Analysis (Lotka Volterra, SIR and Kolmogorov Flow): MG. Visualization: MG. Writing: MG, ST. Writing (Review & Editing): KF, JHM. Funding acquisition: ST, KF, JHM. Supervision: KF, JHM.

ACKNOWLEDGMENTS

We thank all members of the Mackelab, as well as Keisuke Yano and Yoshinobu Kawahara for discussions. MG and JHM are supported by the German Research Foundation (DFG) through Germany's Excellence Strategy (EXC-Number 2064/1, Project number 390727645), the German Federal Ministry of Education and Research (Tubingen AI Center), the "Certification and Foundations of Safe Machine Learning Systems in Healthcare" project funded by the Carl Zeiss Foundation, and the European Union (ERC, DeepCoMechTome, 101089288). MG is member of the International Max Planck Research School for Intelligent Systems (IMPRS-IS). ST and KF are supported by JST CREST JPMJCR2015. ST is supported by Grant-in-Aid for Research Activity Start-up 23K19966 and Grant-in-Aid for Early-Career Scientists 24K20750. KF is partially supported by JSPS Grant-in-Aid for Transformative Research Areas (A) 22H05106.

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

## A    METHODS

### A.1    DERIVATION OF SCORE FACTORIZATION (3)

For $\boldsymbol{x}^{0:T}$ following the simulator $p(\boldsymbol{x}^{0:T}|\boldsymbol{\theta})$, the global score $\nabla_{\boldsymbol{\theta}} \log p(\boldsymbol{\theta}|\boldsymbol{x}^{0:T})$ is given by

$$\nabla_{\boldsymbol{\theta}} \log p(\boldsymbol{\theta}|\boldsymbol{x}^{0:T}) = \nabla_{\boldsymbol{\theta}} \log \left\{ p(\boldsymbol{\theta}|\boldsymbol{x}^{0:T}) \cdot p(\boldsymbol{x}^{0:T}) \right\} = \nabla_{\boldsymbol{\theta}} \log \left\{ p(\boldsymbol{\theta}) \cdot p(\boldsymbol{x}^{0:T}|\boldsymbol{\theta}) \right\}$$

$$= \nabla_{\boldsymbol{\theta}} \log \left\{ p(\boldsymbol{\theta}) \cdot p(\boldsymbol{x}^0|\boldsymbol{\theta}) \prod_{t=0}^{T-1} p(\boldsymbol{x}^{t+1}|\boldsymbol{x}^t, \boldsymbol{\theta}) \right\}$$

$$= \nabla_{\boldsymbol{\theta}} \log \left\{ p(\boldsymbol{\theta}) \cdot p(\boldsymbol{x}^0|\boldsymbol{\theta}) \right\} + \nabla_{\boldsymbol{\theta}} \log \prod_{t=0}^{T-1} p(\boldsymbol{x}^{t+1}|\boldsymbol{x}^t, \boldsymbol{\theta})$$

$$= \nabla_{\boldsymbol{\theta}} \log \left\{ p(\boldsymbol{\theta}|\boldsymbol{x}^0) \cdot p(\boldsymbol{x}^0) \right\} + \nabla_{\boldsymbol{\theta}} \log \prod_{t=0}^{T-1} p(\boldsymbol{x}^{t+1}|\boldsymbol{x}^t, \boldsymbol{\theta})$$

$$= \nabla_{\boldsymbol{\theta}} \log p(\boldsymbol{\theta}|\boldsymbol{x}^0) + \sum_{t=0}^{T-1} \nabla_{\boldsymbol{\theta}} \log p(\boldsymbol{x}^{t+1}|\boldsymbol{x}^t, \boldsymbol{\theta}). \tag{5}$$

Let $\boldsymbol{x}^t$ follow any proposal distribution $\tilde{p}(\boldsymbol{x}^t)$, and $\boldsymbol{x}^{t+1}$ follow the simulation transition $p(\boldsymbol{x}^{t+1}|\boldsymbol{x}^t, \boldsymbol{\theta})$ given $\boldsymbol{x}^t$. Noting that the replacement does not affect the conditional distribution $p(\boldsymbol{x}^{t+1}|\boldsymbol{x}^t, \boldsymbol{\theta})$, we may assume $\boldsymbol{x}^t \sim \tilde{p}(\boldsymbol{x}^t)$ in (5).

By Chain Rule, we have

$$\tilde{p}(\boldsymbol{x}^{t+1}, \boldsymbol{\theta}|\boldsymbol{x}^t) = p(\boldsymbol{x}^{t+1}|\boldsymbol{x}^t, \boldsymbol{\theta}) \cdot \tilde{p}(\boldsymbol{\theta}|\boldsymbol{x}^t) \tag{6}$$
$$= \tilde{p}(\boldsymbol{\theta}|\boldsymbol{x}^{t+1}, \boldsymbol{x}^t) \cdot \tilde{p}(\boldsymbol{x}^{t+1}|\boldsymbol{x}^t),$$

where $\tilde{p}$ means $\boldsymbol{x}^t$ follows the proposal $\tilde{p}(\boldsymbol{x}^t)$. This leads us to the equality

$$p(\boldsymbol{x}^{t+1}|\boldsymbol{x}^t, \boldsymbol{\theta}) = \frac{\tilde{p}(\boldsymbol{\theta}|\boldsymbol{x}^{t+1}, \boldsymbol{x}^t) \cdot \tilde{p}(\boldsymbol{x}^{t+1}|\boldsymbol{x}^t)}{\tilde{p}(\boldsymbol{\theta}|\boldsymbol{x}^t)} = \frac{\tilde{p}(\boldsymbol{\theta}|\boldsymbol{x}^{t+1}, \boldsymbol{x}^t) \cdot \tilde{p}(\boldsymbol{x}^{t+1}|\boldsymbol{x}^t)}{p(\boldsymbol{\theta})}.$$

Here, the second equality is derived from the equality $\tilde{p}(\boldsymbol{\theta}|\boldsymbol{x}^t) = p(\boldsymbol{\theta})$, to which the fact that $\tilde{p}(\boldsymbol{x}^t)$ does not involve a parameter leads. Substituting this equality into Eq. (5), we have

$$(5) = \nabla_{\boldsymbol{\theta}} \log p(\boldsymbol{\theta}|\boldsymbol{x}^0) + \sum_{t=0}^{T-1} \nabla_{\boldsymbol{\theta}} \log \frac{\tilde{p}(\boldsymbol{\theta}|\boldsymbol{x}^{t+1}, \boldsymbol{x}^t) \cdot \tilde{p}(\boldsymbol{x}^{t+1}|\boldsymbol{x}^t)}{p(\boldsymbol{\theta})}$$

$$= \nabla_{\boldsymbol{\theta}} \log p(\boldsymbol{\theta}|\boldsymbol{x}^0) + \sum_{t=0}^{T-1} \nabla_{\boldsymbol{\theta}} \log \tilde{p}(\boldsymbol{\theta}|\boldsymbol{x}^{t+1}, \boldsymbol{x}^t) - T \cdot \nabla_{\boldsymbol{\theta}} \log p(\boldsymbol{\theta}) \tag{7}$$

$$= \sum_{t=0}^{T-1} \nabla_{\boldsymbol{\theta}} \log \tilde{p}(\boldsymbol{\theta}|\boldsymbol{x}^{t+1}, \boldsymbol{x}^t) - (T-1) \cdot \nabla_{\boldsymbol{\theta}} \log p(\boldsymbol{\theta}).$$

Which concludes the proof. Here, the final equality is derived from the assumption $p(\boldsymbol{x}^0|\boldsymbol{\theta}) = p(\boldsymbol{x}^0)$.

### A.2    EXTENSIONS

#### A.2.1    TIME INHOMOGENEOUS SIMULATION

The assumption that $p(\boldsymbol{x}^{t+1}|\boldsymbol{x}^{t+1}, \boldsymbol{\theta})$ does not depend on $t$, referred to as *time-homogeneous* transition simulations, in the main manuscript can be relaxed. Regarding FSNE, the score decomposition (3) also holds for a *time-inhomogeneous* simulators where $p(\boldsymbol{x}^{t+1}|\boldsymbol{x}^{t+1}, \boldsymbol{\theta})$ varies among the time; the score $\nabla_{\boldsymbol{\theta}} \log \tilde{p}(\boldsymbol{\theta}|\boldsymbol{x}^t, \boldsymbol{x}^{t+1})$, however, then depends on time $t$. To address the issue, we train a

time-inhomogeneous score $s_\phi(\boldsymbol{\theta}_a|\boldsymbol{x}^t, \boldsymbol{x}^{t+1}, t)$ using training data $\mathcal{D} = \{(\boldsymbol{\theta}_i, \boldsymbol{x}_i^{t_i}, \boldsymbol{x}_i^{t_i}, t_i)\}$, where $t_i$ is drawn from a random number generator, $\boldsymbol{x}_i^{t_i}$ is drawn from a proposal distribution $\tilde{p}(\boldsymbol{x})$, and $\boldsymbol{x}_i^{t_i+1} \sim \mathcal{T}(\boldsymbol{x}^{t_i+1}|\boldsymbol{x}_i^{t_i}, \boldsymbol{\theta}_i)$. FNPE and FNRE can be handled in the same way as FNLE by adding $t$ as a variable. Experimental evaluations are presented in Appendix E.6.

### A.2.2 Initial Parameter Estimation

The condition on initial state $p(\boldsymbol{x}^0|\boldsymbol{\theta})$ can also be relaxed. Regarding FNSE, the global score $\nabla_{\boldsymbol{\theta}} \log p(\boldsymbol{\theta}|\boldsymbol{x}_o^{0:T})$ can be factorized by

$$\nabla_{\boldsymbol{\theta}} \log p(\boldsymbol{\theta}|\boldsymbol{x}_o^{0:T}) = \nabla_{\boldsymbol{\theta}} \log p(\boldsymbol{\theta}|\boldsymbol{x}_o^0)$$
$$+ \sum_{t=0}^{T-1} \nabla_{\boldsymbol{\theta}} \log \tilde{p}(\boldsymbol{\theta}|\boldsymbol{x}_o^t, \boldsymbol{x}_o^{t+1}) - T \cdot \nabla_{\boldsymbol{\theta}} \log p(\boldsymbol{\theta})$$

as proven in Eq. (7). This implies that the global score is obtained by estimating the two scores $\nabla_{\boldsymbol{\theta}} \log p(\boldsymbol{\theta}_a|\boldsymbol{x}_o^0)$ and $\nabla_{\boldsymbol{\theta}} \log \tilde{p}(\boldsymbol{\theta}_a|\boldsymbol{x}_o^t, \boldsymbol{x}_o^{t+1})$ separately. The estimation of these two scores can be combined by modeling a score function that outputs different scores depending on the number of input variables. FNPE and FNRE can be extended by estimating the two likelihoods $p(\boldsymbol{x}_o^0|\boldsymbol{\theta})$ and $p(\boldsymbol{x}_o^{t+1}|\boldsymbol{x}_o^t, \boldsymbol{\theta})$, similar to FNSE.

### A.2.3 Higher order Markov chains

Our proposal can be generalized to simulators with high order markov chain

$$p(\boldsymbol{x}^{0:T}|\boldsymbol{\theta}) = p(\boldsymbol{x}^{0:m-1}|\boldsymbol{\theta}) \prod_{t=m-1}^{T-1} p(\boldsymbol{x}^{t+1}|\boldsymbol{x}^{t-m+1:t+1}, \boldsymbol{\theta}) \tag{8}$$

with its degree $m$. Under the setting, the score $\nabla_{\boldsymbol{\theta}} \log p(\boldsymbol{\theta}|\boldsymbol{x}^{0:T})$ has the factorization

$$\nabla_{\boldsymbol{\theta}} \log p(\boldsymbol{\theta}|\boldsymbol{x}_o^{0:T}) = \nabla_{\boldsymbol{\theta}} \log p(\boldsymbol{\theta}|\boldsymbol{x}_o^{0:m-1})$$
$$+ \sum_{t=m-1}^{T-m-1} \nabla_{\boldsymbol{\theta}} \log \tilde{p}(\boldsymbol{\theta}|\boldsymbol{x}_0^{t-m+1:t+1}) - (T-m+1) \cdot \nabla_{\boldsymbol{\theta}} \log p(\boldsymbol{\theta}),$$

which implies that the global score can be recovered by the two scores $\nabla_{\boldsymbol{\theta}} \log p(\boldsymbol{\theta}|\boldsymbol{x}^{0:m-1})$ and $\nabla_{\boldsymbol{\theta}} \log \tilde{p}(\boldsymbol{\theta}|\boldsymbol{x}^{t-m+1:t+1})$. Here, $\boldsymbol{x}^{t-m:t+1}$ follows a proposal $\tilde{p}(\boldsymbol{x}^{t-m:t+1})$ and $\boldsymbol{x}^{t-m+1}$ follows the simulation transition $p(\boldsymbol{x}^{t-m+1}|\boldsymbol{x}^{t-m:t+1}, \boldsymbol{\theta})$ in the score $\nabla_{\boldsymbol{\theta}} \log \tilde{p}(\boldsymbol{\theta}|\boldsymbol{x}^{t-m+1:t+1})$. The estimation of these two scores can be merged same as the initial parameter estimation case in the last section.

### A.2.4 Partial Factorization

Instead of considering just a single transition, the methodology naturally extends to multiple transitions, as also explored by Geffner et al. (2023). In PFNSE with $M$, we target $\nabla_{\boldsymbol{\theta}} \log p(\boldsymbol{\theta}|\boldsymbol{x}^t, \dots, \boldsymbol{x}^{t+M})$, while in PFNLE or PFNRE, we target $p(\boldsymbol{x}^t, \dots, \boldsymbol{x}^{t+M}|\boldsymbol{\theta})$ (up to constants), using a dataset of $M$-step simulations. A key advantage of partially factorized methods is that they require fewer network evaluations for inference on a fixed-size observation, which can accelerate sampling and reduce the accumulation of local errors over time. Moreover, longer training simulations potentially lead to superior performance in long-term predictions as fewer local errors can accumulate. However, they may require more simulations for effective training in return for these desirable properties.

### A.2.5 Missing Values in Time Series Observation

Our method is capable of handling missing data. The proposed factorization leverages the Markov properties of time series simulators at observation points, which remain intact even in the presence of missing data. To preserve the amortized property regarding missing points, we can include the time step size $\Delta t$ as an additional input variable to the model. This allows to skip over timepoints that are missing and generally allows for irregular time grids.

## B    SCORE COMPOSITION RULES

Our score decomposition (3) relies on the Markov assumption of the simulator $p(\boldsymbol{x}^{0:T}|\theta)$ as shown in its proof (Appendix A.1). On the other hand, $p(\boldsymbol{x}^{0:T}|\boldsymbol{\theta}_a) = \int p(\boldsymbol{x}^{0:T}|\boldsymbol{\theta})p(\boldsymbol{\theta}|\boldsymbol{\theta}_a)d\boldsymbol{\theta}$ no longer satisfies the Markov property for $a > 0$, which renders Eq. 3 invalid for any $a > 0$. In this section, we review the two existing remedies for tackling this problem.

### B.1    METHOD BY GEFFNER ET AL. (2023)

Geffner et al. (2023) avoids the issue by constructing an alternative sequence of distributions $q_a$ ($0 \leq a \leq A$) that satisfies $q_0 = p(\boldsymbol{\theta}|\boldsymbol{x}^{0:T})$ and $q_A = \mathcal{N}(\boldsymbol{\theta}; \mu_A, \frac{\sigma_A^2}{T}I)$. Adapting their proposal to the Markovian case, we obtain

$$\nabla_{\boldsymbol{\theta}_a} \log q(\boldsymbol{\theta}_a|\boldsymbol{x}^{0:T}) = \sum_{t=0}^{T-1} s_\phi(\boldsymbol{\theta}_a|\boldsymbol{x}^t, \boldsymbol{x}^{t+1}) + \frac{(1-T)(A-a)}{A} \nabla_\theta \log p(\boldsymbol{\theta}).$$

While this approach is generally applicable, it renders the backward SDE (Eq. 2) invalid as the marginals no longer follow the corresponding forward diffusion process. This requires the use of MCMC-based sampling corrections (in the sense of predictor-correct diffusion samplers (Song et al., 2021)), such as unadjusted Langevin dynamics (Geffner et al., 2023) or similar approaches (Sjöberg et al., 2024).

### B.2    METHOD BY LINHART ET AL. (2024)

Linhart et al. (2024) have recently proposed an alternative approach by approximating the marginal scores derived from the forward diffusion process. This enables the utilization of the reverse SDE and consequently allows for the application of standard diffusion-based sampling techniques. They showed that the correct score could indeed be written as suggested in Eq. 3 plus an additional but intractable correction term of the form

$$\nabla_{\boldsymbol{\theta}} \ell(\boldsymbol{\theta}, \boldsymbol{x}_o^{0:T}) = \log \int p(\boldsymbol{\theta}|\boldsymbol{\theta}_a)^{1-T} \prod_{t=1}^{T} p(\boldsymbol{\theta}|\boldsymbol{\theta}_a, \boldsymbol{x}^t, \boldsymbol{x}^{t+1}) d\boldsymbol{\theta}$$

To estimate it analytically, they employed Gaussian approximations,

$$p(\boldsymbol{\theta}|\boldsymbol{\theta}_a, \boldsymbol{x}^t, \boldsymbol{x}^{t+1}) := \mathcal{N}(\boldsymbol{\theta}; \mu_{a,t,t+1}(\boldsymbol{\theta}_a, \boldsymbol{x}^t, \boldsymbol{x}^{t+1}), \boldsymbol{\Sigma}_{a,t,t+1}(\boldsymbol{\theta}_a, \boldsymbol{x}^t, \boldsymbol{x}^{t+1}))$$
$$p(\boldsymbol{\theta}|\boldsymbol{\theta}_a) := \mathcal{N}(\boldsymbol{\theta}; \mu_a(\boldsymbol{\theta}_a), \boldsymbol{\Sigma}_\mathbf{a}),$$

making the additional terms tractable. Eventually, we have

$$\nabla_{\boldsymbol{\theta}_a} \log p(\boldsymbol{\theta}_a|\boldsymbol{x}^{0:T}) \approx \boldsymbol{\Lambda}(\boldsymbol{\theta}_a)^{-1} \left( \sum_{t=0}^{T-1} \boldsymbol{\Sigma}_{a,t,t+1}^{-1} s_\phi(\boldsymbol{\theta}_a|\boldsymbol{x}^t, \boldsymbol{x}^{t+1}) + (1-T)\boldsymbol{\Sigma}_a^{-1} \nabla_{\boldsymbol{\theta}_a} \log p(\boldsymbol{\theta}_a), \right)$$

where $\boldsymbol{\Lambda}(\boldsymbol{\theta}_a) = \sum_{t=0}^{T-1} \boldsymbol{\Sigma}_{a,t,t+1}^{-1} + (1-T)\boldsymbol{\Sigma}_a^{-1}$. This approach requires specifying two "hyper-parameters": the denoising prior covariance $\boldsymbol{\Sigma}_a$ in $p(\boldsymbol{\theta}|\boldsymbol{\theta}_a)$ and denoising posterior covariance $\boldsymbol{\Sigma}_{a,t,t+1}$ in $p(\boldsymbol{\theta}|\boldsymbol{\theta}_a, \boldsymbol{x}^t, \boldsymbol{x}^{t+1})$. The denoising prior covariance is typically known analytically, along with the marginal prior score. Specifically, given a Gaussian prior with covariance $\boldsymbol{\Sigma}_\theta$ we have $\boldsymbol{\Sigma}_\mathbf{a} = (\boldsymbol{\Sigma}_\theta^{-1} + s(a)^2/\sigma(a)^2 \boldsymbol{I})^{-1}$ by Gaussian marginalization rules with the perturbation kernel $p(\boldsymbol{\theta}_a|\boldsymbol{\theta}) = \mathcal{N}(\boldsymbol{\theta}_a, s(a)\boldsymbol{\theta}, \sigma(a)^2 I)$ specific to the diffusion model (i.e. see Appendix Sec. D.2).

In contrast, the denoising posterior covariance must be estimated. To address this, **GAUSS** assumes a Gaussian *clean* posterior (i.e., at $a = 0$) and analytically computes the denoising posterior covariances (analogous to the prior, but with an estimate of $\boldsymbol{\Sigma}_{\boldsymbol{\theta}|\boldsymbol{x}^t, \boldsymbol{x}^{t+1}}$). This estimate can be obtained through samples, given that we have a diffusion model that indeed can sample from $p(\boldsymbol{\theta}|\boldsymbol{x}^t, \boldsymbol{x}^{t+1})$. Alternatively, **JAC** directly estimates the denoising covariance iteratively using Tweedie's Moment projection (Boys et al., 2024), leveraging the Jacobian of the score network:

$$\boldsymbol{\Sigma}_{a,\boldsymbol{x}^t,\boldsymbol{x}^{t+1}} = \frac{m(a)^2}{s(a)^2} \left( \mathbf{I} + \sigma(a)^2 \nabla_{\boldsymbol{\theta}_a} s_\phi(\boldsymbol{\theta}_a|\boldsymbol{x}^t, \boldsymbol{x}^{t+1}) \right)^{-1}$$

If the Gaussian assumption is satisfied, both approaches are theoretically equivalent (neglecting numerical approximation errors). Yet, if these are violated, these yield different approximations to the denoising covariance matrices and hence will behave differently.

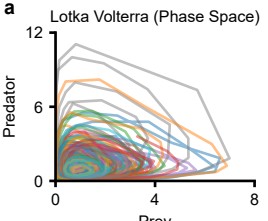 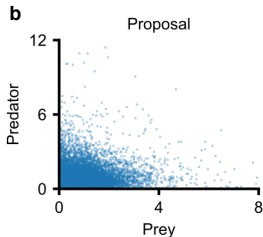 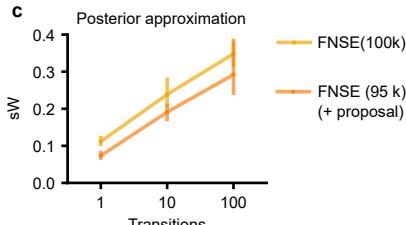

Fig 5: **Proposal**: A set of trajectories within the *phase space* of the Lotka Volterra task for different parameterizations sampled from the prior (constructed using a total of 5k simulation steps) (**a**). The proposal was constructed by randomly sampling noisy points from the state space trajectories (**b**). Posterior approximation in sliced Wasserstein distance constructed by improved proposal compared to our "naively" chosen baseline (**c**).

### B.3 PRACTICAL IMPLEMENTATION DETAILS

The GAUSS and JAC approximation is only valid if $\Lambda$ is positive definite, as the derivation of this form depends on it. Ignoring this can lead to numerical instabilities. Unlike Linhart et al. (2024), we do not clip the diffusion process to a specific prior range to avoid numerical issues; we adapt all estimated posterior precision matrices to ensure positive definiteness of $\Lambda$. This adaptation resolves the problem even for long time series (i.e., $T > 100$; see Appendix E.3 for details). Importantly, since the GAUSS method is derived under Gaussian assumptions, it presupposes that the posterior covariance matrices (and their corresponding scores) are smaller than those of the prior due to observational constraints. This assumption may not hold true, particularly in multi-modal tasks. For instance, in the Periodic SDE task, four modes are symmetrically positioned around the origin. While the variances of each mode may decrease, the total variance often still increases, especially when the modes are distant from the origin. This situation can render $\Lambda(\boldsymbol{\theta}_a)$ non-positive definite, thus invalidating the approximation. Similarly, for JAC, the estimated posterior covariance might not even be positive and definite as the Jacobian of the scoring network might badly represent the Jacobian of the true score.

To address these challenges, we make minimal adjustments to the initially estimated $\boldsymbol{\Sigma}_{a,t,t+1}^{-1}$ to ensure that $\Lambda(\boldsymbol{\theta}_a)$ remains positive definite. We achieve this by considering its eigenvalue decomposition $\Lambda = VAV^{-1}$, allowing us to identify the minimal adjustment required to enforce positive definiteness: $\Lambda_- = -V \min(A, 0) V^{-1}$. Consequently, we update $\hat{\boldsymbol{\Sigma}}_{a,t,t+1}^{-1} = \boldsymbol{\Sigma}_{a,t,t+1}^{-1} + \frac{\Lambda_-}{T} + \epsilon I$ for a small nugget $\epsilon$. The $\Lambda$ computed with these adjusted covariance matrices is guaranteed to be positive definite. The JAC method encounters similar issues, and we apply an analogous adjustment to its initial estimates.

We utilize the GAUSS composition rule as our default method: We estimate the posterior precision matrices $\boldsymbol{\Sigma}_{a,t,t+1}^{-1}$ as recommended by Linhart et al. (2024). For each posterior $p(\boldsymbol{\theta}|\boldsymbol{x}^t, \boldsymbol{x}^{t+1})$, we produce $500 \cdot \dim_{\boldsymbol{\theta}}$ samples in parallel from the diffusion model. These samples are then used to estimate the associated covariance matrix. Although this approach introduces a slight constant overhead prior to each sampling run, it does not require the iterative estimation of Jacobians, as is necessary with the JAC method. On the other hand, within our settings, the dimension of the parameter is often much smaller than the data dimension, rendering computation of the Jacobian not too expensive.

## C PRACTICAL GUIDE TO PROPOSAL CONSTRUCTION

In this section, we provided a practical guideline for constructing the proposal $\tilde{p}(\boldsymbol{x}^t)$. In general, the proposal will be task-specific. For an amortized approximation, a valid proposal $\tilde{p}(\boldsymbol{x}^t)$ should satisfy the conditions: (i) It must have support at any point $\boldsymbol{x}^t$ the simulator can attain (at evaluation time). (ii) The proposed point $\boldsymbol{x}^t$ must be independent of the parameter $\boldsymbol{\theta}$ involved in the transition to $\boldsymbol{x}^{t+1}$ to ensure Eq. 3 remains valid (Appendix Sec. A.1). But solely relying on these properties

| Task | Periodic SDE | | | Linear SDE | | | Mixture RW | | | Double Well | | |
|---|---|---|---|---|---|---|---|---|---|---|---|---|
| | T=1 | T=11 | T=101 | T=1 | T=11 | T=101 | T=1 | T=11 | T=101 | T=1 | T=11 | T=101 |
| FNSE | **0.53** | **0.60** | 0.76 | **0.53** | **0.70** | **0.90** | **0.52** | **0.69** | **0.69** | **0.50** | 0.59 | 0.81 |
| FNSE (pred. prop) | **0.53** | **0.60** | 0.71 | **0.53** | **0.70** | **0.90** | **0.52** | **0.69** | 0.75 | **0.50** | 0.57 | **0.80** |

Table 1: **Predictive proposal:** Posterior approximation for different proposals given as C2ST metric for each benchmark task. Here, we compare our default proposal (trained with 100k step simulations), chosen by domain knowledge, and the prior predictive construction (using 5k simulation steps) training on the remaining 95k step simulations.

| | Lotka Volterra | | | SIR | | |
|---|---|---|---|---|---|---|
| | T=1 | T=11 | T=101 | T=1 | T=11 | T=101 |
| FNSE | 0.57 | 0.78 | 0.94 | **0.67** | **0.78** | **0.74** |
| FNSE (pred. prop) | **0.52** | **0.65** | **0.88** | **0.67** | 0.79 | **0.74** |

Table 2: **Predictive proposal:** Posterior approximation for different proposals given as C2ST metric for Lotka Voletrra and SIR. Here, we compare our default proposal (trained with 100k step simulations), chosen by domain knowledge, and the prior predictive construction (using 5k simulation) trained on 95k step simulations.

does not necessarily make up a "good" proposal. Essentially, the proposal specifies which regions of the data domain are represented in the training dataset. On a finite simulation budget, we thus would expect a better posterior approximation for states that are more likely to be generated by the proposal than those that are less likely. This provides an opportunity and challenge to improve the proposal distribution design for a given simulator.

This interpretation makes it fairly intuitive to know what might be a good proposal and domain knowledge can be used to design a proposal from scratch. If applicants know that the dynamics are bounded (or at least likely contained) within a certain region, then one can design a proposal that roughly covers it. If the applicant knows that the simulator has stationary distributions, then these would serve as good proposals. Yet, in general, it might be hard or suboptimal to construct a proposal in this *naive* way. The default proposals used in our analysis are indeed simple distributions (e.g., Gaussians) chosen based on such knowledge. Yet we here also provide some more general approaches to select reasonable proposals in Sec. C.1 and Sec. C.2.

## C.1 PRIOR PREDICTIVE CONSTRUCTION

Condition (ii) does not generally exclude the use of the simulator to construct a good proposal and we hence can use some simulation outputs to construct a good proposal. As an example, let's consider the Lotka-Volterra task. By sampling from the prior predictive (i.e., sampling parameters from the prior and simulating data), we can visualize the trajectories $x^{0:T}$ of simulation outputs within its phase space (Fig. 5a). These states are likely to encompass most states, and we can use this to construct an effective proposal for our factorized methods, e.g., by simply sampling noisy points on phase space trajectories uniformly (Fig. 5b). For example, one can:

(A) Sample a few parameters $\{\boldsymbol{\theta}_i\}_{i=1}^{N}$ from the prior $p(\boldsymbol{\theta})$.

(B) Perform time series simulations $\boldsymbol{x}_i^{0:T} \sim p(\boldsymbol{x}^{0:T}|\boldsymbol{\theta}_i)$ for each parameters until a certain time $\boldsymbol{x}_i^T$.

(C) Estimate the proposal $\tilde{p}(\boldsymbol{x}^t)$, e.g., by randomly selecting a state $\boldsymbol{x}_i^t$ from $N \cdot (T+1)$ states with some noise (which corresponds to a kernel density estimate).

What condition (ii) prevents is that we cannot directly leverage transition from prior predictive simulations $(\boldsymbol{\theta}_i, \boldsymbol{x}_i^{0:T})$ as each transition will depend on the same parameter, and thus, each triplet $(\boldsymbol{\theta}_i, \boldsymbol{x}_i^t, \boldsymbol{x}_i^{t+1})$ will not satisfy that $\boldsymbol{x}_i^t$ is independent of $\boldsymbol{\theta}_i$. By constructing the proposal as above, apriori, we avoid this problem. Many time series simulators are often designed to ensure that their outputs change minimally for large $T$ (Hayashi et al., 2022; Khatiwala, 2024; Ormeño & General,

2024) (a.k.a. reach stable states or stationarity), a number of simulation trajectories for not too many transitions can, thus, be sufficient to explore the phase space.

Attentive readers might recognize that doing so will itself cost simulations. A quantity we aimed to reduce. To investigate if a better proposal is worth this cost, we tested this on the Lotka Volterra task, comparing it with our default proposal $x^t \sim \text{Unif}(0, 10)$, heuristically chosen by domain knowledge that trajectories will likely oscillate between zero and ten. We equalized the simulation costs for both experiments: 100k simulation steps are used to train the model in the heuristic proposal, whereas, for the prior predictive approach, 5k are allocated to construct the proposal, and 95k for training. The prior predictive construction further improved the performance, even if this cost is subtracted from the training budget (Fig. 5c). We further ran this experiment for all other tasks, which showed that this approach indeed leads to proposals that perform similarly, sometimes better, than the default proposals chosen by us (Tab. 1, Tab. 2). An alternative and related approach, also designed to explore the phase space (but not requiring additional simulations), is the routine applied in the Kolmogorov flow experiment (Appendix Sec. E.5). Here, we simply use a different parameter each time we perform a transition hence avoiding violating condition (ii).

### C.2  TARGET-BASED CONSTRUCTION

We can also create the proposal $\tilde{p}(x^t)$ using any sort of (real/synthetic) set of observations $x_o^{0:T}$ accessible prior to training. For amortized inference we can proceed analogously to Sec. C.1.

If we are only interested in inference for a single observation as in a sequential SBI setting (Papamakarios & Murray, 2016; Greenberg et al., 2019; Sharrock et al., 2024; Deistler et al., 2022), we can construct the proposal using this single observation by simply proposing observed states $x_o^t$ at random. The full amortization property is lost in this way (as the support is constrained), but the simulation budget can be targeted on this more specific inference problem, which can lead to improved performance.

## D  EXPERIMENT DETAIL

### D.1  TASKS

In this section, we elaborate on all tasks used in the analysis in detail. For all tasks, we used a standard normal prior but eventually reparameterized them to certain constraints if necessary for the task (see individual tasks for specification).

Note that the training aims to also amortize over the initial distribution $p(x^0)$. We evaluate the performance always on a specific and fixed initial distribution (see individual tasks for specification), which is also different from the proposal $\tilde{p}(x^t)$.

**Gaussian RW:**  This task serves as a simple baseline. It serves as an extension of the Gaussian Linear task introduced by Lueckmann et al. (2021) to the Markovian case. It is defined by a Gaussian Random Walk of form

$$x^{t+1} = \alpha \cdot x^t + \theta + \epsilon \quad \text{for} \quad \epsilon \sim \mathcal{N}(0, I), \quad \alpha = 0.9,$$

with $x^t, \theta \in \mathbb{R}^d$ and $\theta \in \mathbb{R}^d$. This task offers an analytic solution for the posterior, which is Gaussian. Dimension can be set as wanted, we choose $d = 1, 2, 10$. As a proposal, we choose $\tilde{p}(x^t) = \mathcal{N}(x^t; 0, \sqrt{10}I)$, which is motivated by the fact that it is the variance of the corresponding stationary distribution. For evaluation, we fixed $p(x^0) = \delta(x^0)$ (i.e., a point mass at zero).

**Mixture RW:**  A Mixture of Gaussian Random Walk of form

$$x^{t+1} = x^t + u \cdot \theta + \epsilon \quad \text{for} \quad \epsilon \sim \mathcal{N}(0, I), \quad u \sim \text{Unif}(\{-1, 1\}),$$

with $x^t, \theta \in \mathbb{R}^d$. We choose $d = 2, 5$. As proposal, we choose $\tilde{p}(x^t) = \mathcal{N}(x^t; 0, \sqrt{10}I)$. For evaluation, we fixed $p(x^0) = \delta(x^0)$ (i.e., a point mass at zero).

By design, this task has a tractable transition density, which is a mixture of Gaussians, hence allowing exact computation of the marginal log-likelihood. In contrast to the previous task, this offers a

non-Gaussian bimodal posterior. We generate reference posterior samples using Hamiltonian Monte Carlo (HMC, (Betancourt, 2017)) with 5 integration steps. To avoid mode collapse, we run 100 parallel chains, each initialized on different samples obtained by importance resampling of 50 initial samples from the prior. We run the sampler for 600 burn-in iterations, during which the integration step size is adapted to accept around $80\%$ of the proposed steps. We additionally thin the accepted samples in the chain by a factor of six to avoid autocorrelation.

**Periodic SDE:** The periodic SDE is a linear SDE parameterized as follows:

$$d\boldsymbol{x}^t = \begin{pmatrix} 0 & \theta_2^2 \\ \theta_1^2 & 0 \end{pmatrix} \boldsymbol{x}^t \, dt + 0.1\boldsymbol{I} \, d\boldsymbol{w}^t,$$

which gives us a stochastic oscillator. Each transition corresponds to simulating this SDE for $0.1$ ms, using 20 Euler Maruyama steps. As proposal, we choose $\tilde{p}(\boldsymbol{x}^t) = \mathcal{N}(\boldsymbol{x}^t; \boldsymbol{0}, \boldsymbol{I})$. For evaluation, we fixed the initial distribution to $p(\boldsymbol{x}^0) = \delta((-0.5, 0.5)^T)$, to quickly start with oscillations.

The corresponding marginal log-likelihood is computed using a Kalman filter Kalman (1960). Reference posterior samples are obtained in combination with HMC, using a procedure similar to the one discussed above. This task offers a multimodal posterior (4 modes) with all modes being point symmetric around the origin.

**Linear SDE:** The linear SDE task is given by

$$\boldsymbol{x}^t = (\boldsymbol{A}(\boldsymbol{\theta}) - 2\boldsymbol{I})\boldsymbol{x}^t dt + (0.5\boldsymbol{B}(\boldsymbol{\theta}) + 0.5\boldsymbol{I})d\boldsymbol{w}^t$$

where every entry of $\boldsymbol{A}$ and $\boldsymbol{B}$ is directly parameterized by $\boldsymbol{\theta} \in \mathbb{R}^{18}$ and $\boldsymbol{x}^t \in \mathbb{R}^3$. Reference posterior samples are obtained in the same way as for the previous task. As proposal we choose $\tilde{p}(\boldsymbol{x}^t) = \mathcal{N}(\boldsymbol{x}^t, \boldsymbol{0}, \boldsymbol{I})$. For evaluation, we fixed the initial distribution to $p(\boldsymbol{x}^0) = \delta(\boldsymbol{0})$.

**Double well:** This is a nonlinear SDE

$$d\boldsymbol{x}^t = \theta_1 \boldsymbol{x}^t \, dt + \theta_2 \left(\boldsymbol{x}^t\right)^3 \, dt + \sigma \, d\boldsymbol{w}^t,$$

which samples from a double-well potential with modes position depending on $\theta_1, \theta_2$ (Singer, 2002). As proposal we use $p(\boldsymbol{x}^t) = \text{Unif}(-2.5, 2.5)$, as intial distirbution we at evaluation we use $p(x_0) = \delta((1, -1, 1, -1)^T)$ We use a combination of particle filter (Doucet et al., 2009) and pseudo-marginal-like Metropolis-Hastings MCMC (Andrieu & Roberts, 2009) sampler to generate reference posterior samples.

**Lotka-Volterra:** We use a stochastic Lotka-Volterra model, a classic system of differential equations used to describe predator-prey interactions. We use a stochastic variant, where the amount of noise also depends on the population of each species. The model dynamics are given by the following equations:

$$\frac{dx^t}{dt} = \alpha \cdot x^t - \beta \cdot x^t \cdot y^t dt + \sigma x^t y^t dw_1^t,$$
$$\frac{dy^t}{dt} = -\gamma \cdot y^t + \delta \cdot x^t \cdot y^t dt + \sigma x^t y^t dw_2^t,$$

where $x$ represents the prey population and $y$ represents the predator population. The parameters $\alpha, \beta, \gamma$, and $\delta$ control the interaction between the species. The noise hyperparameter $\sigma$ was set to $0.05$. The dynamic are constrained to remain positive. As a proposal we use $\tilde{p}(\boldsymbol{x}^t) = \text{Unif}(0, 10)$. As evaluation time the initial distribution was fixed to $p(\boldsymbol{x}^0) = \delta(\boldsymbol{x}^0 - (1., 0.5)^T)$.

To sample from the posterior by traditional means, we use a Particle Filter to obtain a stochastic estimator of the marginal likelihood. We use this stochastic estimator in a pseudo-marginal-like Metropolis-Hastings MCMC algorithm to obtain reference samples from the posterior distribution (Doucet et al., 2009; Andrieu & Roberts, 2009).

**SIR:** We use a stochastic SIR method defined through the following equations:

$$\frac{dS^t}{dt} = -\beta S \cdot I + 0.01 dw_1^t,$$

$$\frac{dI^t}{dt} = \beta S \cdot I - \gamma I + 0.1 \cdot I dw_2^t,$$

$$\frac{dR^t}{dt} = \gamma I + 0.01 dw_3^t.$$

here $S$, $I$, and $R$ represent the susceptible, infected, and recovered populations. We use as a proposal $\tilde{p}(\boldsymbol{x}^t) = \text{Unif}(0, 5)$. As an initial distribution, we sample $S^0 \sim \text{Unif}(2.5, 5.)$, $I^0 \sim \text{Unif}(0., 2.5)$ and $R^0 \sim \delta(R^0)$.

To sample from the posterior by traditional means, we use a Particle Filter to obtain a stochastic estimator of the marginal likelihood. We use this stochastic estimator in a pseudo-marginal-like Metropolis-Hastings MCMC algorithm to obtain reference samples from the posterior distribution.

### D.2 TRAINING AND EVALUATION

For the NPE baseline, we utilize a 5-layer neural spline flow (Durkan et al., 2019), with each layer parameterized by a 2-layer MLP having a hidden dimension of 50. Additionally, we employ a Gated Recurrent Unit (GRU) network Cho et al. (2014) as the embedding network, also with a hidden dimension of 50. To enhance the RNN embedding network's ability to generalize across different sequence lengths, we apply data augmentation. Specifically, for each subsequence $T < T_{\max} = 10$, we duplicate $10\%$ of randomly selected parameter-data pairs and shorten the corresponding data time series to length $T$. This does not add any more simulation calls but directly trains the neural net for sequences $T < T_{\max}$.

For FNLE and FNRE, we use adapted reference implementation as implemented in the *sbi* package (Tejero-Cantero et al., 2020b). In FNLE, we use a 5-layer Masked Autoregressive Flow (Papamakarios et al., 2017), each parameterized by a 2-layer MLP with a hidden dimension of 50. In FNRE, we use resnet classifier with two blocks each considering 2 layers with hidden dimension of 50. As MCMC sampling algorithm, we use a per-axis slice sampling algorithm. To avoid mode-collapse, we run 100 parallel chains.

Both approaches are trained with a training batch size of 1000 until convergence, as determined by the default early stopping routine.

For FNSE we use a custom implementation in JAX (Bradbury et al., 2018). In all experiments, we use the Variance Preserving SDE (Song et al., 2021) using

$$f(t) = -0.5 \cdot (\beta_{\min} + t \cdot (\beta_{\max} - \beta_{\min})), \qquad g(t) = \sqrt{\beta_{\min} + t \cdot (\beta_{\max} - \beta_{\min})}$$

We set $\beta_{\min} = 0.1$, and $\beta_{\max} = 10$ for all experiments. Both for the time interval $[10^{-2}, 1.]$. The associated conditional means and variances can be derived from this SDE (Song et al., 2021).

For the score estimation network, we use a 5-layer MLP with a hidden dimension of 50 and GELU activations. The diffusion time is embedded using a random Fourier embedding. Precondition to the scoring network by performing time-dependent z-scoring (Karras et al., 2022). We use the denoising score matching loss with weighting function as in Song et al. (2021).

We use an AdamW optimizer with a learning rate of $5 \cdot 10^{-4}$ with a cosine schedule and a training batch size of 1000. Similar to the SBI routine, we use early stopping, but with a maximum number of 5000 epochs.

## E    ADDITIONAL EXPERIMENTS

### E.1    EXTENDED EXPERIMENTAL COMPARISON

In this section, we extended the empirical evaluation and collected the results in Table 3 and 4. We additionally added the following baseline comparisons:

- NPE' has an alternative embedding network (no RNN). Inspired by the fact that our factorized methods work on all windows ($x^t$, $x^{t+1}$), we embed all such windows into 200 dim feature vectors (using an MLP). All feature vectors are accumulated (using a summation) and mapped into a final 50-dim summary statistics (using another MLP). This is also similar to the embedding networks used in the i.i.d. case (Zaheer et al., 2018).

- $NPE_{50}$ corresponds to our standard NPE with RNN embedding net. By default, we do train NPE on sequences of length ten; here, we extended this to 50 (although this means that the number of joint samples decreases accordingly, i.e., for 10k, only 200 for 100k 2000 to ensure an overall equivalent training budget).

- NPE+ with pre-trained neural sufficient summary statistic. We follow (Chen et al., 2021) infomax learning approach using the distance correlation objective to pre-train the embedding net.

- NPE++ with pre-trained sliced neural sufficient summary statistic (Chen et al., 2023).

- NLE+: NLE with neural sufficient summary statistic (Chen et al., 2021).

- NLE++: NLE with neural sliced sufficient summary statistic (Chen et al., 2023).

- NSE Neural posterior score estimation (Sharrock et al., 2024; Geffner et al., 2023)(analogous to our standard NPE baseline, but with score estimation).

| Task | Periodic SDE | | | Linear SDE | | | Mixture RW | | | Double Well | | |
|---|---|---|---|---|---|---|---|---|---|---|---|---|
| | T=1 | T=10 | T=100 | T=1 | T=10 | T=100 | T=1 | T=10 | T=100 | T=1 | T=10 | T=100 |
| NPE | 0.82 | 0.84 | 0.96 | 0.72 | 0.93 | **0.97** | 0.92 | 0.99 | 1.00 | 0.64 | 0.67 | 0.81 |
| NPE' | 0.78 | 0.91 | 0.99 | 0.70 | 0.93 | 0.99 | 0.91 | 0.99 | 1.00 | 0.55 | 0.72 | 0.99 |
| $NPE_{50}$ | 0.80 | 0.93 | 0.98 | 0.75 | 0.93 | **0.97** | 0.92 | 0.99 | 1.00 | 0.58 | 0.68 | 0.85 |
| NPE+ | 0.80 | 0.89 | 0.99 | 0.75 | 0.92 | **0.97** | 0.91 | 0.99 | 1.00 | 0.67 | 0.65 | 0.91 |
| NPE++ | 0.79 | 0.90 | 0.97 | 0.74 | 0.92 | **0.97** | 0.91 | 0.99 | 1.00 | 0.64 | 0.64 | 0.83 |
| NLE+ | 0.80 | 0.88 | 0.97 | 0.76 | 0.92 | 0.97 | 0.96 | 0.99 | 1.00 | 0.69 | 0.64 | **0.78** |
| NLE++ | 0.75 | 0.93 | 0.98 | 0.76 | 0.93 | **0.97** | 0.95 | 0.99 | 1.00 | 0.55 | 0.60 | 0.81 |
| NSE | 0.77 | 0.92 | 0.97 | 0.68 | 0.93 | **0.97** | 0.91 | 0.99 | 1.00 | 0.60 | **0.59** | 0.79 |
| FNLE | 0.57 | 0.66 | **0.83** | 0.57 | **0.78** | 0.98 | 0.71 | 0.89 | 0.98 | **0.52** | 0.64 | 0.87 |
| FNRE | 0.63 | 0.89 | 0.98 | 0.69 | 0.92 | 0.99 | 0.91 | 0.99 | 1.00 | 0.64 | 0.89 | 0.97 |
| FNSE | **0.56** | **0.65** | 0.85 | **0.56** | 0.82 | **0.97** | **0.57** | **0.78** | **0.92** | **0.52** | 0.68 | 0.89 |
| NPE | 0.68 | 0.63 | 0.95 | 0.70 | 0.84 | 0.96 | 0.91 | 0.99 | 1.00 | 0.62 | 0.64 | 0.76 |
| NPE' | 0.62 | 0.62 | 0.99 | 0.67 | 0.74 | 1.00 | 0.91 | 0.99 | 1.00 | 0.62 | 0.62 | 0.99 |
| $NPE_{50}$ | 0.81 | 0.83 | 0.87 | 0.71 | 0.92 | 0.97 | 0.91 | 0.99 | 1.00 | 0.63 | 0.69 | 0.76 |
| NPE+ | 0.72 | 0.69 | 0.99 | 0.66 | 0.89 | 0.96 | 0.91 | 0.99 | 1.00 | 0.60 | 0.60 | 0.80 |
| NPE++ | 0.67 | 0.65 | 0.95 | 0.71 | 0.91 | 0.97 | 0.91 | 0.99 | 1.00 | 0.65 | 0.61 | 0.78 |
| NLE+ | 0.67 | 0.82 | 0.97 | 0.71 | 0.91 | 0.97 | 0.91 | 0.99 | 1.00 | 0.55 | 0.58 | 0.81 |
| NLE++ | 0.73 | 0.8 | 0.96 | 0.70 | 0.92 | 0.97 | 0.92 | 0.99 | 1.00 | 0.55 | 0.58 | 0.79 |
| NSE | 0.71 | 0.65 | 0.96 | 0.66 | 0.77 | 0.93 | 0.91 | 0.99 | 1.00 | 0.54 | **0.54** | 0.77 |
| FNLE | 0.54 | **0.60** | **0.72** | 0.53 | 0.69 | 0.89 | 0.61 | 0.76 | 0.89 | **0.50** | 0.54 | **0.75** |
| FNRE | 0.55 | 0.75 | 0.92 | 0.54 | 0.81 | 0.99 | 0.70 | 0.86 | 0.93 | **0.50** | 0.74 | 0.90 |
| FNSE | **0.53** | **0.60** | 0.76 | **0.53** | 0.70 | 0.90 | **0.52** | **0.69** | **0.69** | **0.50** | 0.59 | 0.81 |

Table 3: **Extended baseline comparison:** Performance per method given as C2ST metric for each benchmark task. The top half of the table reports results trained on 10k step simulations, and the bottom half on 100k. Best performing methods are marked in bold.

Overall, all "global" methods that aim to reduce the time series into a statically sized summary statistic do behave similarly within our evaluation suite (Table 3, 4). A fundamental challenge with these approaches is sequence length generalization, i.e., finding statistics that generalize to larger lengths than observed in training, which is a challenging task (Ray Chowdhury & Caragea, 2024; Zhang et al., 2022). In contrast, factorized methods do not have this problem. Instead, their main source of error is due to the accumulation of local approximation errors. This can lead to deteriorating performance and is especially visible for NRE within our evaluation. Notably, an increase in C2ST is, to some degree, expected, given that posteriors with more observations likely contract, making slight deviations easier to detect by a classifier. In addition, results are often plotted against the training budget within SBI benchmark (Lueckmann et al., 2017) and not by observation size. We thus visualize our results also in this more similar fashion (see Fig. A6). We

|  | Lotka Volterra | | | SIR | | |
|---|---|---|---|---|---|---|
|  | T=1 | T=10 | T=100 | T=1 | T=10 | T=100 |
| NPE | 0.88 | 0.96 | 0.99 | 0.98 | 0.99 | 1.00 |
| NPE' | 0.84 | 0.96 | 1.00 | 0.97 | 0.99 | 1.00 |
| NPE+ | 0.91 | 0.95 | 0.99 | 0.98 | 0.99 | 1.00 |
| NPE++ | 0.88 | 0.95 | 0.99 | 0.98 | 1.00 | 1.00 |
| NLE+ | 0.92 | 0.96 | 1.00 | 0.98 | 0.99 | 1.00 |
| NLE++ | 0.85 | 0.96 | 0.99 | 0.97 | 0.99 | 1.00 |
| NSE | 0.84 | 0.95 | 0.99 | 0.97 | 0.99 | 1.00 |
| FNLE | 0.75 | **0.89** | **0.98** | 0.93 | 0.96 | 0.96 |
| FNRE | 0.83 | 0.99 | 0.99 | 0.94 | 0.98 | 1.00 |
| FNSE | **0.66** | 0.90 | 0.99 | **0.84** | **0.89** | **0.85** |
| NPE | 0.85 | 0.93 | 0.99 | 0.96 | 0.95 | 1.00 |
| NPE' | 0.83 | 0.87 | 0.99 | 0.98 | 0.99 | 1.00 |
| NPE+ | 0.87 | 0.86 | 0.99 | 0.96 | 0.95 | 1.00 |
| NPE++ | 0.87 | 0.91 | 0.99 | 0.96 | 0.92 | 1.00 |
| NLE+ | 0.87 | 0.92 | 0.99 | 0.96 | 0.96 | 1.00 |
| NLE++ | 0.84 | 0.95 | 0.99 | 0.96 | 0.97 | 1.00 |
| NSE | 0.84 | 0.93 | 0.99 | 0.96 | 0.95 | 1.00 |
| FNLE | 0.60 | **0.78** | 0.96 | 0.93 | 0.95 | 0.97 |
| FNRE | 0.71 | 0.98 | 0.99 | 0.81 | 0.89 | 0.93 |
| FNSE | **0.57** | **0.78** | **0.94** | **0.67** | **0.78** | **0.74** |

Table 4: **Extended baseline comparison for Lotka Volterra and SIR**: Performance per method given as C2ST metric for Lotka Volterra and SIR task. The top half of the table reports results trained on 10k step simulations, and the bottom half on 100k. Best performing methods are marked in bold.

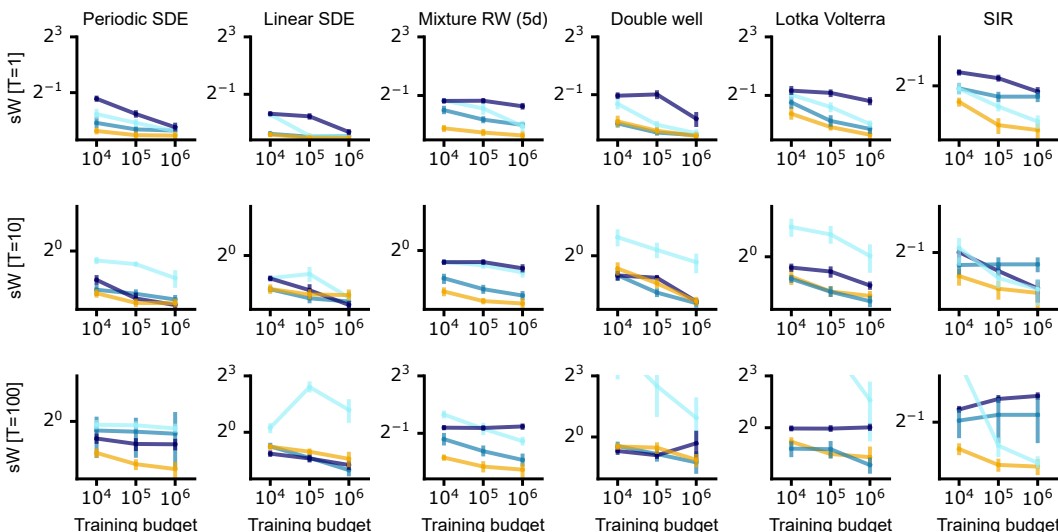

Fig 6: **Posterior approximation plotted against training budget**: Each column corresponds to a task. Each row shows the performance in sliced Wasserstein distance for test observations generated using 1,10, and 100 transitions. In contrast to the main figures, we see the x-axis now specifies the training budget, i.e., the number of transition evaluations that can be used to simulate the training data.

performed this on a training budget (equaling the number of transition step evaluations) of $10^4$, $10^5$, and $10^6$ (which corresponds to $10^3$, $10^4$, and $10^5$ "full" simulation for NPE) as in Lueckmann et al. (2021).

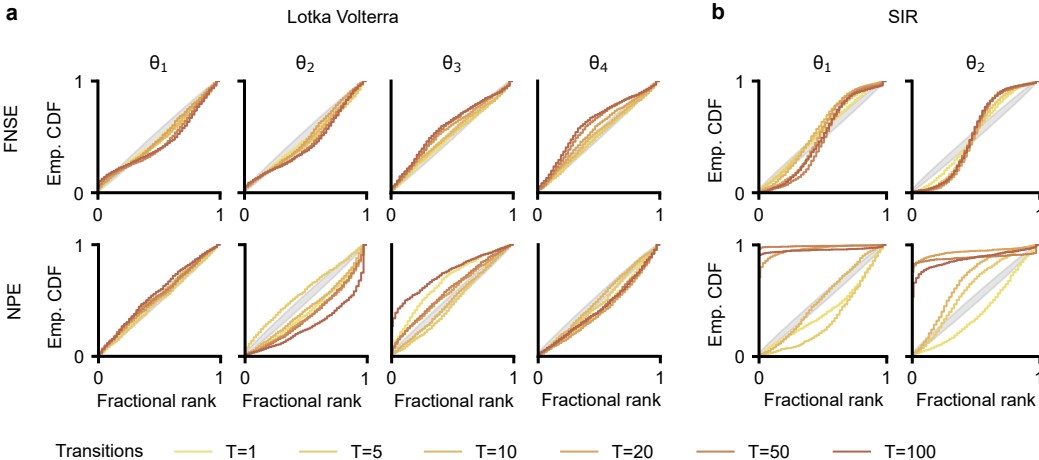

Fig 7: We perform a simulation-based calibration analysis (SBC) (Talts et al., 2018) for the Lotka Volterra and SIR task trained using 100k step simulations using FNSE and NPE. We used 1000 test simulations for evaluation. The shaded Grey area indicates the expected results under the true posterior. Colors indicate the number of transitions of the test simulations.

## E.2 CALIBRATION ANALYSIS LOTKA VOLTERRA AND SIR

We performed a simulation-based calibration analysis (Talts et al., 2018) for the Lotka-Volterra and SIR tasks. Consistent with our evaluation using C2ST and sW, we observe that FNSE is better calibrated than NPE across different sequence lengths (Fig. 7ab). This difference is especially pronounced in the SIR task, where NPE completely fails to generalize to sequence lengths larger than those observed during training ($T > 10$). In contrast, the effect is less prominent in the Lotka-Volterra task. We hypothesize that, due to the periodic nature of the Lotka-Volterra dynamics, the embedding network more easily identifies representations that generalize well across sequence lengths.

## E.3 COMPOSITION METHODS

We compared the results of the different implemented score composition methods. Overall this showed a clear advantage of our implementations of GAUSS/JAC over FNPE (Fig. 10). Even if Langevin corrections are introduced FNPE does tend to perform worse and diverge given a larger number of observations.

The performance of both the GAUSS and JAC methods is quite similar (Fig. 10, first column), especially in the Gaussian RW task. This is expected as both are equivalent if Gaussian assumptions are not violated. Notably, for samplers with fewer steps on non-Gaussian tasks, JAC tends to perform slightly better than GAUSS on long time series (Fig. 10**b**, columns two onwards). The imposed regularity conditions on the posterior covariances appear to resolve the numerical challenges associated with this approach, as observed by Linhart et al. (2024).

## E.4 PROPOSAL DISTRIBUTION

The choice of proposal distribution can have a significant impact on a finite simulation budget. In our benchmark tasks, we primarily selected simple yet reasonable distributions based on our prior knowledge of the dynamics involved. However, we examine the impact of this choice more closely in the Gaussian RW and Mixture RW tasks (see Fig. 11).

We set the proposal distribution to a Gaussian with a zero mean and variable variance, ranging from very narrow to quite wide. Importantly, at evaluation, we initialize the distribution as a point mass at zero. For a very narrow proposal, we thus anticipate strong performance at $T = 1$. However, as $T$ increases, performance may decline because the time series dynamics could extend beyond the

high-probability support of the proposal. Conversely, excessively wide proposals may also hinder performance if the values at the observation time fall outside the training set. It is crucial to note that this performance evaluation is relative to our specific evaluation pipeline. In both tasks, evaluations are conducted on random walks that consistently start from zero, thus constraining the range of attainable values given a maximum number of steps.

Overall, there is indeed an optimal proposal for all tasks, which is close to, but not identical to, our intuitively chosen options. As expected, the significance of the proposal also increases with dimensionality.

It is important to emphasize that the proposals we utilized are generally not optimal. Other tasks, such as the Periodic/Linear SDE, Double Well, SIR, and Lotka-Volterra models, exhibit a high correlation among different variables, which the proposal we chose does not capture. See Appendix Sec. 3.4 for some examples. Nevertheless, to ensure a fair comparison with the NPE baseline, we opted for relatively simple proposals.

### E.5 KOLMOGOROV FLOW

For the Kolmogorov flow, we modify the score architecture. We utilize a convolutional embedding network composed of four blocks, each containing a Conv2D layer, GroupNorm, and GeLU activation. The output is then processed by a two-layer MLP to produce a 100-dimensional embedding. This embedding is generated from the two frames passed to the score network and subsequently concatenated. On this embedding, we employ the same five-layer MLP that we use for all other tasks, but we increase the hidden dimension to 400.

It is important to note that in the NPE approach, we would need to construct an embedding network not only for individual "images" but also for a complete video of arbitrary length, which poses significant challenges.

As an initial distribution, we utilize a filtered velocity field provided by `jax_cfd` library, with a maximum velocity and peak wave number of three. Our approach closely follows the methodology outlined by Rozet & Louppe (2023). The simulator performs 100 solve steps per transition, with a step size of $10^{-3}$ seconds; each transition thus emulates 10 milliseconds. To introduce stochasticity into the PDE, we add small amounts of Gaussian noise with a standard deviation of $5 \cdot 10^{-3}$ after each transition. We used the `jax-cfd` library (Kochkov et al., 2021) to solve the Navier-Stokes equations on a $64 \times 64$ grid.

Designing an effective proposal distribution in a high-dimensional setting is challenging. Simply using the initial distribution is unlikely to yield satisfactory performance, as the dynamics evolve over time. Ideally, we would like to obtain samples from $p(\boldsymbol{x}^t)$ for a range of different $t$; however, this approach would require a significant simulation budget to step to each $t$.

To address this challenge while still generating a diverse set of simulations, we employ the following scheme:

(i) Sample an initial value $\boldsymbol{x}^0 \sim p(\boldsymbol{x}^0)$

(ii) For $t = 0, \ldots, T$:
 – Sample $\boldsymbol{\theta}_i \sim p(\boldsymbol{\theta})$
 – Perform a transition $\boldsymbol{x}_{t+1} \sim \mathcal{T}(\boldsymbol{x}^{t+1}|\boldsymbol{x}^t, \boldsymbol{\theta}_i)$

Notably, this procedure does not violate the requirement that $\tilde{p}(\boldsymbol{x}^T)$ is independent of $\boldsymbol{\theta}$, as each transition is performed with a different, independent parameter. This approach is motivated by the fact that, at best, we would like proper samples from the associated prior predictive distribution (Appendix Sec. C).

For 1,000 initializations, we run this procedure for 100 steps, and for 10,000 initializations, we run it for 10 steps, totaling exactly 200,000 transition emulations.

To assess the average performance across different observations and observation lengths, we conducted a simulation-based calibration analysis (Talts et al., 2018) (see Fig. 12). This analysis evaluates whether the true parameter ranks across posterior samples are uniformly distributed, as would be expected under the true posterior distribution. In other words, it examines whether the estimated

posteriors adequately cover the true parameters. The results indicate a generally good, though not perfect, calibration of the estimated posteriors. However, the calibration tends to deteriorate with longer observation lengths. In this scenario, the posterior becomes very narrow, making the metric sensitive to such narrow distributions (which increases the likelihood of missing the true posterior). Furthermore, this suggests that for certain observations, the approximate posterior may be biased. Given the limited amount of training data, such outcomes are to be expected.

### E.6 TIME-INHOMOGENOUS GAUSSIAN RANDOM WORK

To evaluate the performance of the time-inhomogeneous extension described in Appendix A.2.1, we consider the Time-Inhomogeneous Gaussian Random Walk (TI Gaussian RW)

$$\boldsymbol{x}_{t+1} = \left(\alpha + \frac{1}{t+1}\right) \cdot \boldsymbol{x}_t + \boldsymbol{\theta} + \epsilon \quad \text{for} \quad \epsilon \sim \mathcal{N}(0, \boldsymbol{I}),\, \alpha = 0.9,\, \boldsymbol{x}_t, \boldsymbol{\theta} \in \mathbb{R}^d$$

with $\boldsymbol{x}_t$ multiplied by the time-inhomogeneous coefficient $\left(\alpha + \frac{1}{t+1}\right)$. We demonstrate the scalability of our methods with dimension $d$ ranging from $1, 2$, to $10$, and the number of observation transitions ranging from $1, 10$, to $100$. As a comparison metric, we use C2ST. The simulation budget is fixed at $10k$ and $100k$, consistent with the problem settings in the main section. Note that the training data include the time variable $t$, and therefore, the training data for estimating the transition distribution $p(\boldsymbol{x}^{t+1}|\boldsymbol{x}^t, \boldsymbol{\theta})$ are reduced to $\frac{1}{100}$ compared to the time-homogeneous setting in the main section. The maximum time length $T_{\max}$ in training is set to $100$.

The experimental results are presented in Fig. 13. We can confirm that the proposed methods outperform the NPE baseline overall. In particular, it has been confirmed that FNSE scales better for longer observation times compared to other methods.

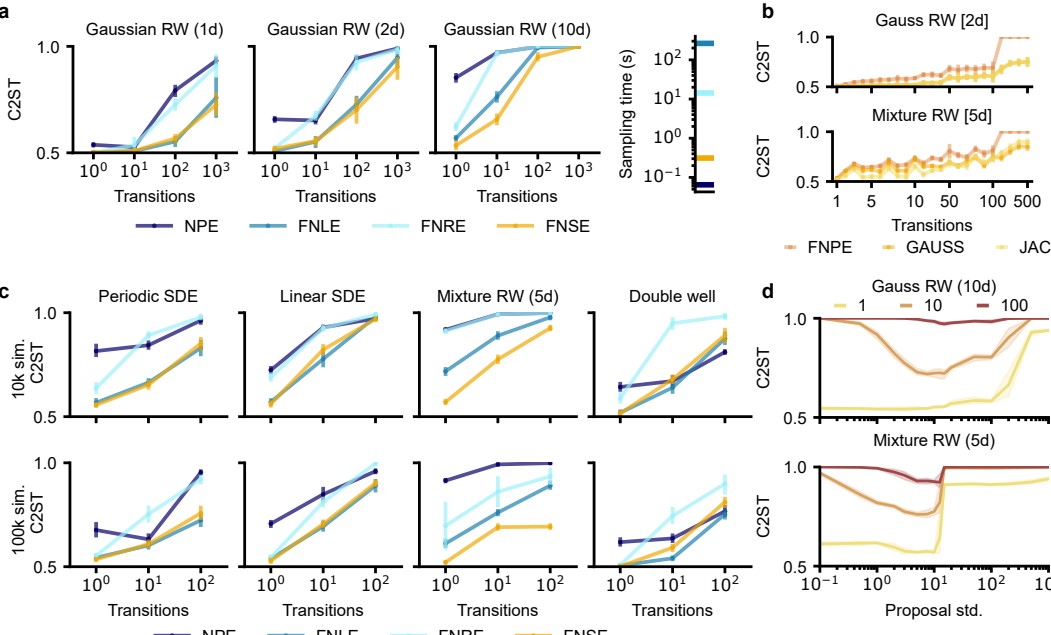

Fig 8: **Benchmark with C2ST metric:** We validate our method on a Gaussian Random Walk with different dimensions for different lengths (i.e. Transitions), also tracking sampling times (**a**). We assess FNSE score accumulation over Gaussian RW and Periodic SDE tasks using a fixed Euler–Maruyama sampler (**b**). We compare methods across tasks and transition steps (**c**). Finally, we examine the effect of the proposal on NFSE trained with 10k simulations from a normal proposal of varying standard deviation (**d**).

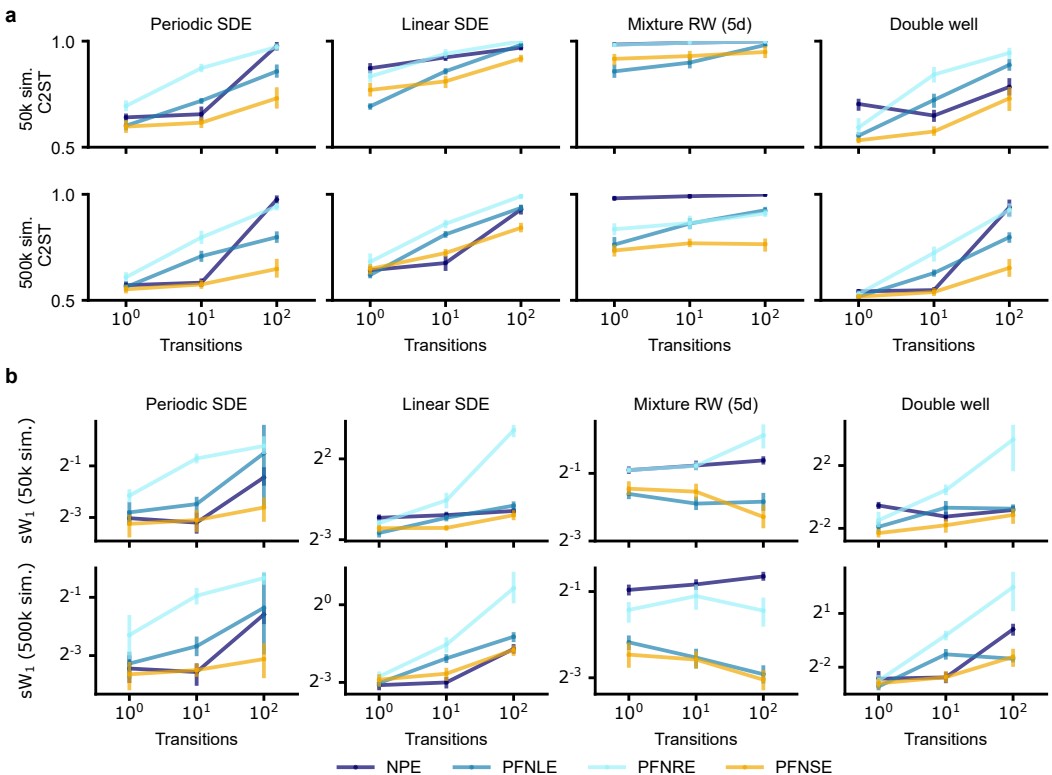

Fig 9: **Benchmark partially factorized methods:** We show the benchmark performance using partially factorized methods PFNLE, PFNRE, and PFNSE. Each is trained with 5 steps. The performance is shown with respect to C2ST (**a**) and sliced Wasserstein distance (**b**).

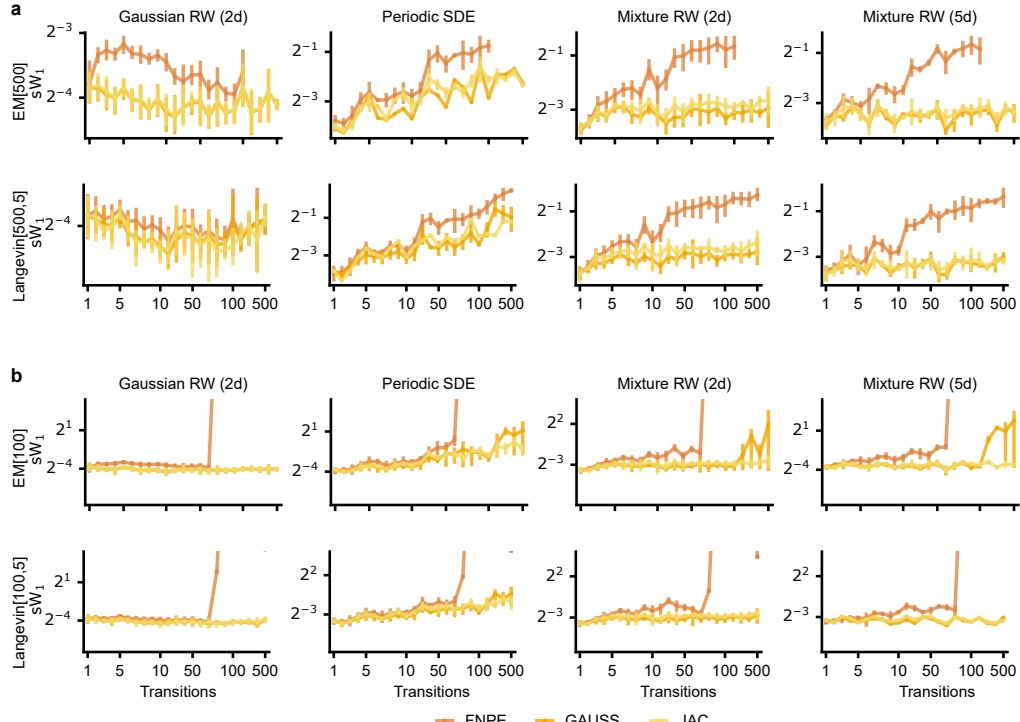

Fig 10: **Benchmark score composition methods:** Performance of score estimators trained with 10k simulations using different diffusion samplers (default and fewer steps) and score composition methods. Each column corresponds to a different task, and each row represents either a standard Euler-Maruyama diffusion sampler or one equipped with a Langevin corrector, both performing 5 steps. The analysis is conducted for two time discretizations: (**a**) 500 steps and (**b**) 100 steps.

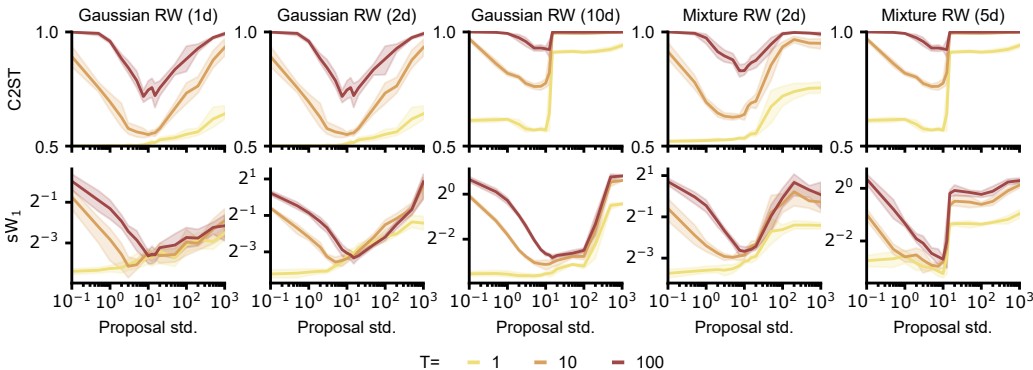

Fig 11: **Benchmark proposal sensitivity:** Each column represents a different task. The top row displays results using the C2ST metric, while the bottom row shows the sliced Wasserstein distance. FNSE models are trained with a fixed budget of 10k simulations, with each using different proposal distributions characterized by a mean of zero and varying standard deviations (x-axis).

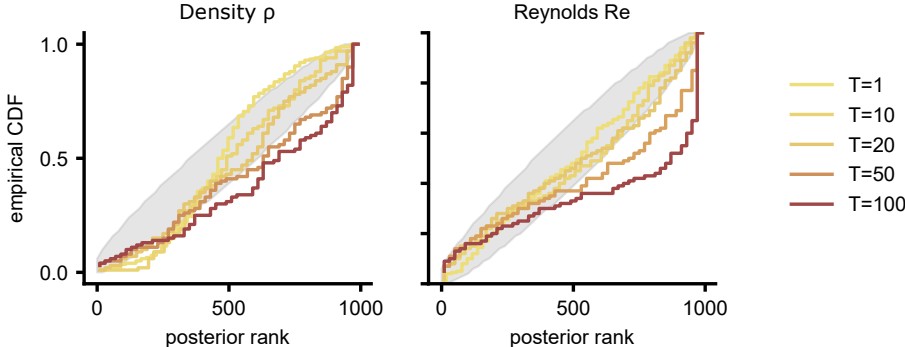

Fig 12: **Simulation-based calibration Kolmogorov flow**: The simulation-based calibration results present the empirical cumulative density functions (CDF) of the ground-truth parameters, ranked according to the inferred posteriors derived from 100 different observations. A well-calibrated posterior should exhibit uniformly distributed ranks, as highlighted by the shaded gray area. Repeated for data emulated for T=1,10,20,50,100 steps).

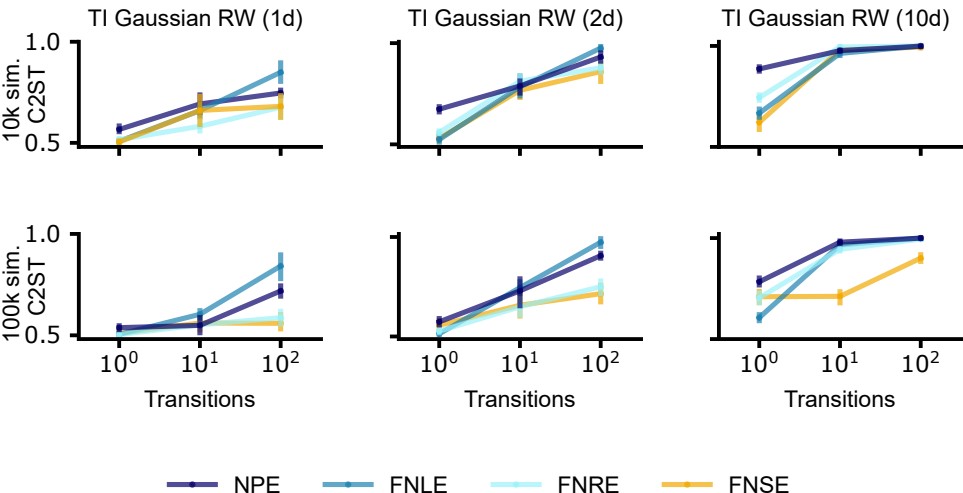

Fig 13: **Time inhomogenous Gaussian random walk**: We train all methods on 10k (top row) and 100k (bottom row) simulation on the time-inhomogenous Gaussian RW talk. All methods have also been adapted to input the current time (as given by integers ranging from zero to 100).

