# OpenReview forum: "Compositional simulation-based inference for time series"
_ICLR.cc/2025/Conference — ICLR 2025 Poster_

### Official Review · Reviewer_MmYK · 2024-10-15

**Soundness:** 3
**Presentation:** 3
**Contribution:** 2
**Rating:** 6
**Confidence:** 2

**Summary:**

The authors consider performing simulation-based inference for Markovian simulators. They propose two primary methods, based on score matching, and likelihood (ratio) estimation. In both cases, they utilize the Markovian structure to increase the efficiency of inference, by performing inference using data from single-state transitions, rather than requiring whole sequence simulation. In the likelihood case, the global likelihood is the product of the transition functions $p(x^{t+1}|x^t, \theta)$ (eq. 1), which once estimated can be used with MCMC to infer the posterior. Similarly, for estimating the score, they again use the Markovian structure to factorize the score (eq. 3), and use a "local" score estimator to approximate the global score. In both cases, a proposal distribution is used, $\tilde{p}(x^t)$, from which a sample is drawn, and a single transition is performed, which is used for learning.

**Strengths:**

The method is to the best of my knowledge novel and provides a solution to utilizing the known dependency structure of Markovian simulators; knowledge which is often neglected in simulation-based inference.  The application area is an important area and handling sequence data in simulation based inference is known to be often challenging (in terms of performance and computational cost). The experiments are convincing and consider a set of familiar but interesting models.

**Weaknesses:**

My main criticism is that the paper should include a clearer description of what might constitute a "good" proposal distribution $\tilde{p}(x^t)$, especially when it is first introduced. I further feel that the requirement for choosing this proposal is a major problem for the practical utility of the introduced methods, and the authors should better address this problem. E.g. should we aim for it to resemble a prior predictive simulation for a randomly chosen $t$? Could the proposal $\tilde{p}(x^t)$ be sequentially improved as the posterior is learned? The use of hand-picked simple distributions is somewhat concerning for broader applicability.

Some smaller issues:
- The names of the contributed methods, at least FNSE, should be introduced earlier, for example, in the final paragraph of the introduction (FNSE is in Figure 1, but it's easy to miss the name).
- "except FNRE; in LV": I don't believe LV abreviation was introduced, presumably Lotka-Volterra.
- Text size in figures is often a little too small.
- There is a reasonable degree of entangling of previous work in section 3.2.2, which made it hard to read. For example, the abbreviation FNPE is introduced without definition, initially implied to refer to the method by Geffner. It is then introduced and thereafter used as an adaptation of the Geffner method. It also refers to a score-based method, so the use of NPE is confusing as it implies a neural posterior estimation method, using e.g. flows.

**Questions:**

Is $x^t$ in the SIR example the three states $\{S^t, I^t, R^t\}$? Correct me if I am misunderstanding, but if this is the case, the proposal is  $\tilde{p}(x^t) = \text{Unif}(0, 5)$, presumably independent for for each state S, I and R. At any given time $t$, I would expect the states to be correlated (e.g. more recovered likely implies less infected and susceptible individuals). The use of an independent uniform does not capture this. Further, I presume the method is only valid  if we observe all three states?

As mentioned above: Should we aim for $\tilde{p}(x^t)$ to resemble a prior predictive simulation for a randomly chosen $t$? Could the proposal $\tilde{p}(x^t)$ be sequentially improved as the posterior is learned?

---

> ### Author Response · Authors · 2024-11-22
> **Response to reviewer MmYK**
>
> We thank the reviewer for the constructive feedback and for recognizing the novelty and importance of our work. We address each of your concerns in detail below and outline the steps we will take to improve clarity and practical applicability in response to your review. We also thank the reviewer for pointing out smaller issues within the manuscript, which we will address in the updated version of the manuscript.
>
> ### On Weakness:
>
> > “My main criticism is that the paper should include a clearer description of what might constitute a "good" proposal distribution $p(x_t)$, especially when it is first introduced… Should we aim for p~(xt) to resemble a prior predictive simulation for a randomly chosen t?”
>
> We thank the reviewer for their valuable feedback. The performance of the proposed method is to some degree, sensitive to the selection of proposals, as shown in Fig. 2b and Fig. 8 (where we ablate over a large number of different Gaussian proposals). We agree that the current manuscript does not sufficiently discuss their construction. In response to your feedback,  we have provided a practical guide to proposal construction (Sec. 3 in [here](https://filebin.net/mr0aicst9v8p44hb)). This will also be included in the revised manuscript, and we aim to improve our presentation in Sec. 3.
>
> ### On Questions:
>
> > “Is xt in the SIR example the three states St,It,Rt? Correct me if I am misunderstanding, but if this is the case, the proposal is p~(xt)=Unif(0,5), presumably independent for each state S, I and R. At any given time t, I would expect the states to be correlated …”
>
> Indeed, as the reviewer recognizes, the proposal we use is non-optimal (but by choice). Proposals that align with task correlation structure are expected to perform better. We decided to use a “simple” proposal across all benchmark tasks to demonstrate that our method also performs well in these cases, given that good proposals are not necessarily trivial to obtain (without using simulations themselves!). Although, it can be worth spending some simulations to construct a good proposal (see [here](https://filebin.net/mr0aicst9v8p44hb), Sec. 3).
>
> > "Further, I presume the method is only valid if we observe all three states?"
>
> Yes, we require Markovness. Marginalizing the state can lead to non-Markovian simulators.
>
> > *“Could the proposal p~(xt) be sequentially improved as the posterior is learned?...”*
>
> Yes. Sequential methods in SBI usually target only a posterior for a specific observation $x_o^{0:T}$, whereas we here aim for an amortized approximation. Notably, a good proposal for a specific observation (or multiple ones) is to simply sample $x_o^t$ for random t (as at inference time, we will evaluate this observation). The proposal $p(x^t)$ can be freely updated over time in the sequential setting. However, the best way to improve it is not totally clear. An interesting direction for future work could be to update the proposal based on the Fisher information provided by $x_o^t$ (per time), which would avoid wasting simulations in regions that are uninformative for the $\theta$ (and tractable given that in FNSE, the networks estimate the score).

---

> > ### Comment · Area_Chair_oXvM · 2024-11-26
> >
> > Dear reviewer,
> >
> > Please make to sure to read, at least acknowledge, and possibly further discuss the authors' responses to your comments. Update or maintain your score as you see fit.
> >
> > The AC.

---

> > ### Comment · Reviewer_MmYK · 2024-11-26
> >
> > Thank you for the clarifications. Although the additional experiments are somewhat limited (e.g. for a specific task), the results look good, and the authors have added some discussion regarding my primary concern. I have adjusted my rating from a 5 to a 6.

---

> > > ### Author Response · Authors · 2024-11-27
> > >
> > > We thank the reviewer for reevaluating our work and are glad we addressed the primary concern. We have extended the analysis of the proposal to the other tasks within the revised manuscript, which shows that the construction performs similarly, or better than our default proposal (Appendix C).

---

### Official Review · Reviewer_Ck63 · 2024-10-23

**Soundness:** 3
**Presentation:** 2
**Contribution:** 2
**Rating:** 6
**Confidence:** 4

**Summary:**

This work proposes a set of Simulation-Based Inference (SBI) methods to perform approxiamte Bayesian (parameter) inference given an
observation that takes the from of a specific class of timeseries (finite-length homogenous Markov chains).
These methods are adaptations of existing score, likelihood, and ratio-based estimation methods, whose original formulation
assume i.i.d observations, and not observations arising from a Markov Chain.

The methods do so by estimating ratio/likelihood/scores "locally", e.g. to the individual factors of the Markov chain.
They then produce a "global" posterior estimate (e.g. an estimate of the posterior of the parameters given the entire Markov chain sample)
by relying on the factorization structure of the Markov chain probability. The composition is straightforward in the case of ratio and likelihood methods,
while when using scores, additional approximations are necessary due to the non-Markovian structure of the blurred posteriors estimated in the previous step.

These approximations are direct adaptations of the one already used in prior work:
- one performs iterative Langevin-based sampling of an annealing path of distributions bridging from a standard Gaussian distribution to the true posterior, and whose intermediate distributions are precisely the SDE solutions whose scores were estimated.
- one produces an approximation of the scores of the SDE initialized at the posterior of the parameter given the entire Markov trajectory, which can then be used to simulate trajectories from the time-reversed SDE, whose final iterate is marginally distributed according to the true posterior.

The performance of the method is investigated on a set of experiments, which include standard benchmark models and a Kolmogorov flow model with high dimensional observations. The simulators are adapted (using noise injection) to fit the formalism of stochastic inference methods.

**Strengths:**

The paper proposes a conceptually simple, attractive adaptation of SBI methods to Markov chains. For a total simulation budget, the shift towards learning the transition factor allows to increase the number of available observations. Moreover, learning the transition probabilities may be a simpler problem than learning the parameter-to-whole-timeseries relationship, as the observation is lower dimensional in the former case.

**Weaknesses:**

**Regarding Presentation**

The presentation of the method needs some improvements.

First, regarding the the sampling part, the presentation of FNPE and Gauss do not define precisely key details of the approximations.
- In FNPE, where is $q_a$ used for instance?
- In GAUSS/JAC, What is $\Sigma_{a,t, t+1}$?

I had to look at each of the separate papers that introduced such approximations in the iid case to understand what was going on.
The paper should be updated to define all terms properly -- most of will constitute background. Even though the adaptations made by the method are almost independent from the these terms, the paper should remain self contained as far as the core method is concerned. To save space, the FNPE approximation method could be placed in the appendix, as it is not used in the experiments.

Second, the local score estimation section could also be made clearer: for instance, this section
starts by breaking down the true (unblurred) posterior (Equation (3)), highlighting how knowing the score of each
factor is enough to get the score of the resulting posterior. However, what is actually needed for sampling is the blurred posterior,
which cannot be factorized as Equation (3). Equation (3) thus ends up not being very relevant to the method.


**Regarding experiments**: the presentation of the Kolmogorov flow experiments is missing details: for instance, it does not include any mention of stochasticity, which is confusing given that the method is about estimating probability distributions.


**Regarding performance**
- The main weakness of the method right now is that the number of timesteps investigated in the paper remains smaller (up to 100)
compared to the regime of some application of interest, like neuroscience.
- In the Kolmogorov flow experiments, the MAE of the posterior predictive significantly increases for when using 100 instead of 10. It would be nice if the authors commented on this point in the paper.

**Questions:**

Is there a particular motivation behind using Sliced Wasserstein Distance for evaluation? If using it as the main criterion, it would be nice to provide an intuition as of what features of differences in distribution this distance is capturing. Otherwise, I'd suggest using the Sinkhorn Divergence instead -- with a small regularization parameter, it is essentially a direct approximation of the Wasserstein distance.

**Minor**

- The use of "time-variant" and "time invariant" should be replaced by the more standard "(time) homogenous/unhomogenous".
- "MCMC-based sampling corrections, such as unadjusted Langevin dynamics". The ULA doesn't include a correction step (it is "undajusted"). What do the authors mean by that?
- To produce symmetric quotation marks, consider using ``text'' instead of "text" in latex.

---

> ### Author Response · Authors · 2024-11-22
> **Response to reviewer Ck63**
>
> We thank the reviewer for your thoughtful review and constructive feedback. We are glad that you found our approach to simulation-based inference (SBI) for Markov chains conceptually appealing and well-aligned with the aim of increasing efficiency through local transition factor learning. We address each of your points below and outline the steps we will take to improve clarity and expand upon the technical and practical details in the revised manuscript.
>
> ### On Weaknesses:
>
> >  “The presentation of the method needs some improvements…. should remain self-contained ... .To save space, the FNPE approximation method could be placed in the appendix, as it is not used in the experiments…”
>
> Thank you, for pointing out this issue in clarity. Our goal was to provide a concise overview, but we understand that this made it difficult to follow without referring to prior works. We will follow your suggestions and adapt the manuscript accordingly:
> - We will move parts from FNPE to the appendix and expand on GAUSS/JAC
> - We will adapt the corresponding sections to be self-contained.
>
> > “The presentation of the Kolmogorov flow experiments is missing details”
>
> We thank the reviewer for pointing this out. We will add missing details and move important details from the appendix into the main section.
>
> > “The main weakness of the method right now is that the number of timesteps investigated in the paper remains smaller (up to 100),...”
>
> We want to point out that previous work in the i.i.d. case only used up to 30 i.i.d. samples in their empirical evaluation (Geffner et al. 2022, Linhart et al. 2024). We acknowledge that for time series, it is even more relevant to support larger lengths. We chose the limit of 100 when comparing all the tasks, as we require ground-truth reference samplers for each task within the main evaluation, and not all samplers scale efficiently and robustly beyond. For the tasks where analytic or efficient sampling algorithms are available, we have also extended the analysis to 500 or 1000 transitions (Fig 2ab, Appendix Fig. 7). We aimed to address the performance degradation for FNSE over more and more timesteps (till 500) and diffusion solvers in Appendix Fig. 7 within the original manuscript. The performance did not degrade strongly over time in these tasks using GAUSS or JAC (in contrast to FNPE).
>
> > “In the Kolmogorov flow experiments, the MAE of the posterior predictive significantly increases for when using 100 instead of 10. It would be nice if the authors commented on this point in the paper”
>
> The increase is naturally attributed to the natural divergence of the time series over time (due to noise and different parameters). We will add this explanation to the manuscript.
>
> ### On Question:
>
> > “"MCMC-based sampling corrections, such as unadjusted Langevin dynamics". The ULA doesn't include a correction step (it is "unadjusted"). What do the authors mean by that?”
>
> We do mean ULA. In the context of diffusion models, samplers improved using MCMC methods like ULA, which are often referred to as predictor-corrector methods (where ULA would be the corrector). We will clarify this.
>
> > “Is there a particular motivation behind using Sliced Wasserstein Distance for evaluation … Otherwise, I'd suggest using the Sinkhorn Divergence instead.”
>
> The main motivation was that sliced Wasserstein distances were also used in related prior work (Linhart et al. 2024). Furthermore, it is “parameter-free” in contrast to MMD (kernel) and Sinkhorn (regularization strength) and can be computed very efficiently (making it, in our eyes, a good metric). C2ST is a more common metric within related literature, is saturated, and is very sensitive for narrow distributions, making it ill-suited to show relative performance differences.
> Slicing preserves the metric axioms and the weak continuity of the divergence, implying that the sliced divergence will share similar topological properties (Kimia Nadjahi, 2022). At the same time, we agree that Sinkhorn divergence would be a good metric and might be more interpretable for people familiar with optimal transport. Yet, we think there is no large enough benefit to providing this additionally (or instead).

---

> > ### Comment · Area_Chair_oXvM · 2024-11-26
> >
> > Dear reviewer,
> >
> > Please make to sure to read, at least acknowledge, and possibly further discuss the authors' responses to your comments. Update or maintain your score as you see fit.
> >
> > The AC.

---

> > > ### Comment · Reviewer_Ck63 · 2024-11-28
> > > **Thanks**
> > >
> > > I thank the reviewers for their efforts in clarifying the manuscript. I'm OK with keeping the SW distance.
> > >
> > > > The increase is naturally attributed to the natural divergence of the time series over time (due to noise and different parameters)
> > >
> > > OK - are the plotted metrics showing an average error  (normalizing by the number of timepoints). If so, is it fair to say that the average performance gets worse as T increase? I am just trying to get a sense of how well the method currently works for longer time series.

---

> > > > ### Author Response · Authors · 2024-12-02
> > > >
> > > > We thank the reviewer for their evaluation and interest in our work. Sorry for the misunderstanding; it is the average error across samples from the posterior predictive (not accounting for time length, but normalized across the number of predictive samples). In addition, the boxplots show the distribution of fifty distinct observations from which our method estimated posterior samples, which are simulated to posterior predictive samples, and then the absolute error to the observation is computed. So, even if we plugin the true parameter, the simulation on large T will diverge from the true observation due to intrinsic noise in the simulation (even without noise, slight deviations of the parameters (as present in the samples) will have a similar effect). We have improved the description of it in the figure caption upon the original submission (but we did not highlight it as a major revision).
> > > >
> > > > ### Regarding on long time series in general:
> > > >
> > > > The accumulation of local approximation errors in factorized methods can lead to worse approximations on long-term observations. All of our experiments show that this issue is especially prominent for FNRE but only modestly for FNLE and FNSE. Other methods that rely on embedding nets or summary statistics (like NPE) will also inherently fail on too long time series as, in general, as there does not exist a finite-dimensional sufficient summary statistic. Furthermore, the embedding network might not generalize beyond the (bounded) training observation length. The new Appendix, Fig 6, illustrates it fairly well. Specifically, given more training data, our methods also improve for all investigated time lengths (T=1,10,100).
> > > > In contrast, the NPE baseline mostly improves only for sequence lengths within its training domain (T<=10), but this improvement does not translate to T=100 on most tasks. This is somewhat expected as, given a perfect local approximation, we would also expect a perfect posterior approximation on arbitrary long-time series. In practice, this does not mean that any factorized method will generalize to a very long time series; rather, it scales more efficiently than the competing baselines. Overall, we agree with the reviewer that one should expect the average performance to decline when increasing $T$ (as also present in our results), similar to any method that operates similarly (e.g., numerical ODE solvers).
> > > >
> > > > We do want to point out that in the manuscript, we mostly focus on learning from single-step simulations (an extreme end of a spectrum, Sec. 3.2.1 line 213). On very long observations, single-step factorized methods can also be expensive, as for estimating the global target, we need to evaluate the local neural network for each step (see Fig. 2a, which is especially prominent for FNLE). On the other hand, we can extend the approach to, e.g., five steps instead of a single one (“partially factorized”, as discussed in Sec. 3, A.2.4 and results in Appendix Fig. 9). Particularly at inference time, we thus only have to consider all windows of size five that overlap by one (if the Markov degree is one). Although this might require more training budget (see Appendix Fig. 9), this can be preferable for a very long time series: it is more efficient (the need for fewer network evaluations when estimating the global target) and can reduce the accumulation of local errors.

---

### Official Review · Reviewer_bjS2 · 2024-10-31

**Soundness:** 3
**Presentation:** 3
**Contribution:** 2
**Rating:** 6
**Confidence:** 4

**Summary:**

The authors introduce a novel approach for simulation-based inference (SBI) for time series models of which the joint probability density is Markovian. The work builds on the recently introduced framework of score-based generative models for SBI which aims to approximate the score of the posterior distribution when multiple data points (in this case time points) are conditioned on. They show that their approach is beneficial over a state-of-the-art method in time series benchmarks.

**Strengths:**

- The paper proposes an intuitive approach for SBI for (very) high-dimensional time series.
- The topic (SBI for high-dimensional time series data) is very timely and relevant given the (seemingly) increased interested of applied researchers.
- The paper is well written and easy to follow.

**Weaknesses:**

- Despite the intuitive idea, the proposed method is very incremental. As far as I can see, instead of modelling the score $s(\theta_a|x^t)$ the paper proposes to model $s(\theta_a|x^t, x^{t-1})$, i.e., adding one variable to the score network input, in order to make the method amendable for time series?
- The evaluations are in my opinion not fully convincing.
    - The authors chose to compare their approach to exactly one other baseline which I find a bit insufficient given the wealth of SBI methods and the fact that the authors propose approaches for local FNLE, FNRE and FNSE.
    -  The baseline NPE uses an RNN as an embedding network. Given the authors assume Markovianity, they could have, e.g., just computed summary statistics and used this as conditioning variables, instead of running a costly RNN (which also needs to be trained in addition to the flow layers). Alternatively approaches that reduce the dimensionality more efficiently like neural sufficient statistics could have been benchmarked for a more thorough evaluation [1,3].
  - Relatedly, there is work on high-dimensional SBI that the authors could potentially have included in their evaluations: [1-4]

#### References
- [1] Generalized massive optimal data compression, 2017, https://arxiv.org/abs/1712.00012
- [2] Neural Approximate Sufficient Statistics for Implicit Models, 2021, https://arxiv.org/abs/2010.10079
- [3] Is Learning Summary Statistics Necessary for Likelihood-free Inference?, 2023, https://proceedings.mlr.press/v202/chen23h.html
- [4] Simulation-based inference using surjective sequential neural likelihood estimation, 2023, https://arxiv.org/abs/2308.01054

**Questions:**

- Is there any reason why NPE performs so badly in the C2ST statistic in Figs 3c and 3d? Previously literature, e.g., [1-2], seems to report different results.
- Could you kindly explain the exact experimental setup in Fig2a and Fig2c? Does, e.g., $100$ transitions mean that NPE received $100$ data samples of length $100$ while for $10$ transitions it received $1000$ samples?

#### References
- [1] Flow Matching for Scalable Simulation-Based Inference, Figure 4, https://arxiv.org/pdf/2305.17161
- [2] Benchmarking Simulation-Based Inference, Figure 2, https://arxiv.org/abs/2101.04653

---

> ### Author Response · Authors · 2024-11-22
> **Response to reviewer bjS2**
>
> We thank the reviewer for the detailed review. We are glad that the reviewer found our approach intuitive, timely, and well-presented. In the following, we will address the reviewer's concerns and clarify aspects of our work. Below, we respond to each of your points individually.
>
> ### On Weaknesses
> >  *"The proposed method is very incremental"*.
>
> We recognize that our approach draws upon recent advances in score composition (Geffner et al., 2023; Linhart et al., 2024), and is based on an simple insight (which, furthermore, we maintain is novel!) -- with this conceptually simple change, we can substantially increase the scope of applicability of this approach. Almost all mechanistic simulators of time-series models are Markovian. Simulation-based inference for time-series models has been highly challenging, and our insight provides a general and effective way of addressing this issue! We do not agree that the effect of this insight, and thus the contribution of the study, is incremental!
> Our contribution lies in demonstrating how, based on our observation,  these techniques can be adapted and effectively utilized for simulation-based inference in Markovian simulators – a “timely and significant task”. We demonstrate that posteriors can be efficiently approximated using only single (or few) step simulations. To our knowledge, this approach is novel and relevant, and addresses an important problem.
>
> > “The evaluations are in my opinion, not fully convincing”
>
> The reviewer points out that we only compare our method to "exactly one baseline," which they consider "a bit insufficient given the wealth of SBI methods." In response, we have expanded our experimental evaluation to include additional SBI methods. Specifically, we have incorporated Neural Score Estimation (NSE) as a baseline (as well as approaches based on neural summary statistics, see below). Furthermore, we thank the reviewer for highlighting related literature, and we have included several of these methods in our evaluation to provide a more comprehensive comparison.
>
> The reviewer pointed out that we could have “just computed summary statistics and used this as conditioning variables”. We want to clarify that, for our experimental evaluation using NPE, we require summary statistics that receive a sequence of variable size, compressing it into a static vector (in a likelihood-free setting, [1] would itself require estimating the likelihood ahead of time). This statistic should not only be nearly sufficient but also generalized to a “long” time series (larger than observed within training). We want to point out that this is a general and open problem in sequence models (e.g. [E1]). We do provide a sensible, commonly used baseline using an RNN (see, e.g., [E2]). Finding ``the best’’ approach lies outside the scope of the study. But we agree that an additional embedding approach would be useful to comparison and included it, which performed similar as our original baseline (see [here](https://filebin.net/mr0aicst9v8p44hb), NPE’ Sec 2.).
>
> We thank the reviewer for pointing out the work on “neural sufficient statistics”, and agree that it would be a valuable addition to our experimental validation. Specifically, we included Neural sufficient statistics [3], based on the Infomax criterium using the distance correlation approximation and Sliced Neural sufficient statistics [2]  (see [here](https://filebin.net/mr0aicst9v8p44hb), N(L/P)E+(+) Sec 2.). These baselines could sometimes improve upon our original NPE baseline, but overall, they performed similarly and distinctively from the factorized versions.  In addition, we will extend our related work section to include [1-4]. We will incorporate these new extended results in Appendix C within the next day.
>
> [E1] Zhou, Y., Alon, U., Chen, X., Wang, X., Agarwal, R., & Zhou, D. (2024). Transformers can achieve length generalization but not robustly. arXiv preprint arXiv:2402.09371.
>
> [E2] Dyer, J., Cannon, P. W., & Schmon, S. M. (2021, June). Deep signature statistics for likelihood-free time-series models. In ICML Workshop on Invertible Neural Networks, Normalizing Flows, and Explicit Likelihood Models.

---

> ### Author Response · Authors · 2024-11-22
> **Response bjS2 (part II)**
>
> ### On Questions
>
> > “Is there any reason why NPE performs so badly in the C2ST statistic in Figs 3c and 3d? “
>
> NPE results in Fig 3c seem to align with Lueckman et al. results (C2ST ~ 1.). However, while the tasks are inspired by Lueckman et al. 2022, there are some key differences that explain the discrepancy:
> - The considered tasks are different. As Markov models, they are implemented through a stochastic differential equation (SIR additionally has a stochastic initial condition). Both involve a state-dependent diffusion term (state-correlated noise) which makes the task more challenging. Observations also differ (details can be found in Appendix B.1).
> - Our simulation budget is reported based on the number of “transition steps.” Depending on how many “transition steps” per full simulation are required, the simulation budget for other work must be adjusted accordingly.
>
> > “Could you kindly explain the exact experimental setup in Fig2a and Fig2c? Does, e.g., 100 transitions mean that NPE received 100 data samples of length 100 while for 10 transitions it received 1000 samples?”
>
> Within our evaluation, the training budget (equaling the number of transition steps) is fixed across methods to, e.g., 10k. This means:
> - Factorized methods obtain 10k pairs $(\theta, x^t, x^{t+1})$ as training data.
> - NPE using 10-step simulation, in contrast, receives 1k paris of $(\theta, x^{0,10})$. Totaling to exactly the same number of required transitions. In addition, this dataset is augmented with $(\theta, x^{0,i})$ for $i<10$ to promote generalization across sequence lengths (without additional simulations).
>
> In Fig. 2a and Fig2c, we evaluate the corresponding methods on sequences generated using $T=1,10,100$ transitions while the training budget of steps is fixed to 10k or 100k transition evaluations.

---

> > ### Comment · Area_Chair_oXvM · 2024-11-26
> >
> > Dear reviewer,
> >
> > Please make to sure to read, at least acknowledge, and possibly further discuss the authors' responses to your comments. Update or maintain your score as you see fit.
> >
> > The AC.

---

> ### Comment · Reviewer_bjS2 · 2024-11-27
>
> Thank you very much for addressing my concerns and questions.
>
> After having read the revision to the paper and the other reviews, I am willing to raise my score to 6.
> The technical contribution, which I still find to be fairly incremental, prohibits a higher score for me.

---

> > ### Author Response · Authors · 2024-11-27
> >
> > Thank you for taking the time to reconsider our manuscript. We are pleased to hear that our revisions have addressed your main concerns and questions.

---

### Official Review · Reviewer_Z2NW · 2024-11-02

**Soundness:** 4
**Presentation:** 4
**Contribution:** 3
**Rating:** 8
**Confidence:** 4

**Summary:**

The paper proposes a class of methods to perform neural simulation-based inference on time-series data when the forward model has Markovian structure, i.e. when the next step only depends on the current one. This is the case in many physical systems defined by PDEs, for example. The method exploits this structure of the simulator / forward model to learn locally (in timestep) amortized estimators for global parameters of interest that describe the system in question. The paper exploits many recent advances in amortized simulation-based inference and adapts them to the Markovian time-series setting.

**Strengths:**

- The paper is exceptionally well-written. Secs. 2 and 3 set up the problem well, and describe the contributions in the context of existing literature. I feel like I learned a lot about the surrounding field beyond the specific contributions of the paper by reading these sections. Limitations are discussed upfront, and some extension settings explored in an initial way.
- The experiments in the paper are well-motivated. The paper looks at toy models, standard benchmarks, as well as more challenging simulators that highlight some of the specific advantages of the method (e.g., generalization beyond the number of steps used during training).
- The paper explores a timely question of broad scientific as well practical real-world interest. The generalization capabilities of the method beyond the number of timesteps trained for seem especially practical (and maybe should be highlighted more) -- many applications are bottlenecked by the inability to practically train on the longer-horizon timescales needed during deployment.

**Weaknesses:**

While the core methods are described nicely, some practical aspects receive less thorough treatment. Specifically, the proposal distribution seems like a critical design choice (especially for complex problems like Kolmogorov flow), and the score composition rules prove surprisingly robust even when their assumptions are violated (e.g. GAUSS in non-Gaussian setting). There is limited discussion of why, and limited principled guidance on these implementation choices that practitioners might need to apply the method to new domains. A more thorough discussion here could significantly strengthen the paper in particular for practitioners across domains.

**Questions:**

- How sensitive is the more complex Kolmogorov flow example to the choice of proposal distribution?
- Beyond the empirical results demonstrated, do the authors have a deeper understanding of the robustness of score composition rules?
- The generalization beyond the number of timesteps seen is one of the most useful properties of the method -- how do the generalization capabilities depend on the number of timesteps used during training (e.g., max 10 vs 100 timesteps during training, then 1000 during evaluations)?

---

> ### Author Response · Authors · 2024-11-22
> **Response to Z2NW**
>
> We thank the reviewer for the positive evaluation of our manuscript and the constructive feedback. The reviewer found the paper “exceptionally well-written” and that it provided valuable insights into the field. We appreciate your recognition of the strengths of our work, especially the practical relevance and the generalization capabilities of our method. We address your comments and questions below.
>
> ### On Weakness:
>
> > *“ Specifically, the proposal distribution seems like a critical design choice”*
>
> We agree that the current manuscript does not sufficiently discuss the proposal's construction. In response to your variable comment, we have provided the two practical and general approaches to construct the proposal (Sec. 3 in [here](https://filebin.net/mr0aicst9v8p44hb)), and we will revise our submission to include them.
>
> > the score composition rules prove surprisingly robust even when their assumptions are violated.
>
> We can confirm that the Gaussian correction consistently improves performance, a conclusion also supported by Linhart et al. (2024). However, this finding only demonstrates that the correction enhances performance relative to the version without it; it does not imply that the Gaussian approximation perfectly represents the true correction term. As discussed in Section 5.2, some room for improvement remains, particularly in relaxing the Gaussian assumption, which represents an important direction for future research.
>
>
>
> ### On Questions:
>
> > “How sensitive is the more complex Kolmogorov flow example to the choice of proposal distribution?”
>
> The selection of the proposal distribution is important, especially in such structured high-dimensional simulations. It should be designed to ensure that its support encompasses possible simulation outputs over time (see answer on the proposal), and our construction in Appendix C.4 aims to address this in a simulation-efficient manner.
>
> > “Beyond the empirical results demonstrated, do the authors have a deeper understanding of the robustness of score composition rules?”
>
> The Gaussian approximative assumptions in both GAUSS and JAC are only used within approximating the “correction term”, which involves the “denoised posterior” $p(\theta|\theta_a,x)$. While its influence is less significant at the end of the diffusion process (near noise), it becomes important once we approach the data distribution (Appendix C, Linhart et al. 2024). Intuitively, in this region, if now e.g., the posterior is multimodal, the distribution $p(\theta|\theta_a,x)$ is approximately not (as $\theta_a$ is $\theta$ + low amounts of noise so that the modes will collapse around the location of $\theta_a$) and can thus be to some degree approximate by a Gaussian. Of course, this cannot be claimed for larger noise levels.
>
> Anyway, a big reason for preferring the Gaussian approximation is its simplicity and computational tractability. The method by Linhart et al. 2024 can, for example, be analogously extended with  Mixtures of Gaussians (under similar assumptions). Yet, at least a naive application will run into computational challenges that must be addressed approximately (products of Gaussian will stay a Mixture of Gaussian but with exponentially increasing components, which in our case would be time steps).
>
> > “The generalization beyond the number of timesteps seen is one of the most useful properties of the method -- how do the generalization capabilities depend on the number of timesteps used during training (e.g., max 10 vs 100 timesteps during training, then 1000 during evaluations)?”
>
> Longer training simulations are expected to enable the training of score networks based on extended states, potentially leading to superior performance in long-term predictions.  But instead, it incurs significant simulation costs during training. I will include this trade-off between prediction performance and simulation cost in Appendix of the revision.

---

> > ### Comment · Reviewer_Z2NW · 2024-11-22
> >
> > Thanks for the thoughtful responses! I maintain that this is a good paper, after also looking at the other reviewer comments, and keep my accept score.

---

> > > ### Author Response · Authors · 2024-12-02
> > >
> > > Thank you very much for your prompt response and for recommending acceptance.
> > >
> > > We sincerely appreciate your thoughtful engagement with our work and the detailed comments, which have significantly improved our submission.

---

### Official Review · Reviewer_g3YX · 2024-11-03

**Soundness:** 2
**Presentation:** 2
**Contribution:** 2
**Rating:** 5
**Confidence:** 5

**Summary:**

The paper applies compositional score-matching, previously introduced for SBI on exchangeable models, to non-exchangeable, Markovian models. The appealing aspect of the methodology is that a score estimator can provide good posterior estimates without simulating the full process for each batch instance, but instead simulating single transitions per batch instance and learning to aggregate across time. The hope is that one can achieve the same posterior accuracy with less than $N * T$ evaluations, where $N$ is the simulation budget and $T$ is the maximum time horizon of the simulator.

**Strengths:**

- The paper is self-contained, easy to follow, and it attempts to address a challenging problem that is relevant to the field as a whole.
- The method is applicable to different classes of SBI methods (e.g., likelihood approximation, likelihood ratio approximation, direct posterior estimation).
- The idea is original and can stimulate further research into efficient SBI on dynamic models, especially when simulation budgets are scarce.

**Weaknesses:**

- As far as I understand it, the proposed method is a straightforward extension of FNPE with an additional input to the score estimator (Eq.6 in [1]). As such, the claim that a new “general SBI framework” is proposed requires some calibration. In contrast, the authors could highlight and extend the empirical aspects of the work.

- While the basic idea is rather appealing, I find it a bit unconvincing that FNPE can robustly approximate the correct global posteriors with sparse training (the same goes for FNLE). There is an aspect of the methodology that strikes me as magical: suppose that most information about $\theta$ in a time-varying signal is contained in later time segments (e.g., as in certain non-stationary signals) or subject to latent transitions (e.g., as in HMMs). It would be incredibly hard to get to that information if the simulation is not evolved for longer $T$, but the factorized approach is supposed to somehow get enough training signal from very sparse simulations in all relevant time windows. How is it possible for the score network to properly learn the correct composition without any assumptions on signal stationarity or smoothness? I assume the reasonable performance on the toy examples can be attributed to the rather short signal lengths and the use of simple, low-dimensional models (Fig.9 reveals striking miscalibration for longer time series on the only challenging model, confirming my fears).

- Overall, the evaluation lacks robustness and I am concerned that it is subject to randomness given the extremely limited number of test simulations used (10!). Since the work is explicitly situated in an amortized inference context and pitched as such, the evaluation could have been much more comprehensive, featuring hundreds or even thousands of test simulations and at least some practically relevant metrics with an absolute interpretation, such as calibration error or other measures used for evaluating computational faithfulness in Bayesian analysis ([2]). In addition, there are no ablation studies (e.g., one such study could vary the simulation budget or use an adaptive solver for better-than-uniform sampling, another study can consider extremely long time horizons $T$), all models only have a few parameters (SIR and LV are toy models from the 60s, but presented as distinct from the toy model section), and it is hard to tell if the performance on the only non-trivial model (4.4) is acceptable given the lack of ground-truth and the poor calibration for not not even that long time horizons ($T=100$).

- It may be helpful to clarify some points in Section 2.2 regarding related work on amortized inference. It might strengthen the section to highlight that amortized Bayesian inference is explicitly introduced in [3] and extensively discussed and expanded upon in inference compilation methods [4, 5, along with additional references therein]. Including this line of research could offer a more comprehensive overview of the field. Additionally, [6, 7] don't seem to introduce, discuss, or validate amortized methods; instead, they focus on sequential likelihood-free techniques, such as SNPE [7], without suggesting or claiming to use amortization. In fact, [6] cursorily note that learning over the prior predictive is possible but ultimately dismiss it as “grossly inefficient” (p.3).

[1] Geffner, T., Papamakarios, G., & Mnih, A. (2023, July). Compositional score modeling for simulation-based inference. In International Conference on Machine Learning (pp. 11098-11116). PMLR.

[2] Gelman, A., Vehtari, A., Simpson, D., Margossian, C. C., Carpenter, B., Yao, Y., ... & Modrák, M. (2020). Bayesian workflow. arXiv preprint arXiv:2011.01808.

[3] Gershman, S., & Goodman, N. (2014). Amortized inference in probabilistic reasoning. In Proceedings of the Annual Meeting of the Cognitive Science Society (Vol. 36, No. 36).

[4] Le, T. A., Baydin, A. G., & Wood, F. (2017). Inference compilation and universal probabilistic programming. In Artificial Intelligence and Statistics (pp. 1338-1348). PMLR.

[5] Wu, M., Choi, K., Goodman, N., & Ermon, S. (2020). Meta-amortized variational inference and learning. In Proceedings of the AAAI Conference on Artificial Intelligence.

[6] Papamakarios, G., & Murray, I. (2016). Fast ε-free inference of simulation models with bayesian conditional density estimation. Advances in Neural Information Processing Systems, 29.

[7] Lueckmann, J. M., Goncalves, P. J., Bassetto, G., Öcal, K., Nonnenmacher, M., & Macke, J. H. (2017). Flexible statistical inference for mechanistic models of neural dynamics. Advances in Neural Information Processing Systems, 30.


*Minor*
Some attention to typos and neologisms is needed (e.g., P2L107 -> one can sample, …, to its probability density?, P7L377, etc.).

**Questions:**

Why does sW seem to, counterintuitively, increase for increasing number of transitions (Figure 2)?
Will the method be robust to missing or corrupted data?

---

> ### Author Response · Authors · 2024-11-22
> **Response to reviewer g3YX**
>
> We thank the reviewer for the detailed and extensive review. We appreciate that the reviewer finds our paper “self-contained” and “easy to follow,” addressing a “relevant” problem with an “original” idea.
>
> ### On Weakness
>
> >  *“the proposed method is a straightforward extension of FNPE”*
>
> Our approach draws upon recent advances in score composition (Geffner et al., 2023; Linhart et al., 2024), but we do not regard this as a weakness-- rather,  our contribution lies in noticing how these techniques can be adapted (in a conceptually simple manner) to substantially improve simulation-based inference for Markovian simulators of time series. We show that posteriors can be efficiently approximated using only single (or few) step simulations. This approach is novel and effective. We do agree that the wording of “general SBI framework” should be tempered, and have modified this in the manuscript (we now call it “approach to SBI in time series”).
>
> > *“I find it a bit unconvincing that FNPE can robustly approximate the correct global posteriors with sparse training. … suppose that most information about θ in a time-varying signal is contained in later time segments (e.g., as in certain non-stationary signals) or subject to latent transitions (e.g., as in HMMs).  It would be incredibly hard to get to that information if the simulation is not evolved for longer T, but the factorized approach is supposed to somehow get enough training signal from very sparse simulations in all relevant time windows. How is it possible for the score network to properly learn the correct composition without any assumptions on signal stationarity or smoothness?”*
>
> We want to clarify that this study focuses specifically on Markov simulators (Hidden Markov Models (HMM), which the reviewer mentions are beyond the scope of this work, as noted in “Limitations” (Section 5.2.)).  To clarify the scope further and highlight the inclusion of significant simulators, we will expand Section 1 with additional explanations and illustrative examples. However, for Markovian simulators, inference can be performed using just pairwise transitions by composing scores as we describe!
>
> For example, let's assume that we have a Markov process in which most information about the parameters is contained in long-term simulation from a specific initial distribution ($T >> 0$), as in the example by the reviewer. Our approach can still leverage this information if the proposal shares sufficient support with the marginal distributions at $T$. We agree with the reviewer that having stationarity is beneficial for our methodology; for example, the stationary distribution would serve as a good proposal in the above scenario. For an unstable process, any ML-based approach will inevitably struggle given that, at some point, trajectories will simply fall out of training distribution (for too large T). Our approach is not excluded from this problem; however, for any fixed maximum $T$, the proposal can be designed to avoid this problem (similar to the maximum $T$ in training and other methods). We will aim to clarify this in the revised manuscript.
>
> > *“Overall, the evaluation lacks robustness and I am concerned that it is subject to randomness given the extremely limited number of test simulations used (10!)”.*
>
> We carefully checked the reviewer's concern by reevaluating parts of the benchmark using 100 test simulations, which did not lead to qualitative changes but confirmed and strengthened our initial results (see [here](https://filebin.net/ffcadasjgnpf7bo7), Sec. 1). We also want to clarify that we not only visualize the performance over 10 test simulations, but we plot the performance over five independent runs (training + evaluation), each considering different test simulations. This approach is in line with related studies (Lueckmann et al. 2021, Dax et al. 2023, Sharrock et al. 2024) that use 10 test samples but only one run (e.g. Geffner et al. 2023 use 6 test samples for five independent runs, Linhart et al. 2024 uses 25 test samples for one independent run).
>
> > *“...at least some practically relevant metrics with an absolute interpretation”*
>
> We used metrics that are commonly used in the literature, and we do maintain that our current suite of metrics is sufficient to demonstrate our claims and, indeed, has an absolute interpretation (0 in sW and 0.5 in C2ST correspond to perfect point-wise approximations).
>
> We do, however, agree with the reviewer that a calibration analysis would strengthen our manuscript, and we hence performed an additional analysis on the Lotka Volterra and SIR task (see [here](https://filebin.net/ffcadasjgnpf7bo7) Sec. 4). The results are in line with our original benchmark, verifying that our methods are better calibrated than the NPE baseline (although not perfectly across time steps as also indicated by imperfect sW/C2ST due to limited training budget). We will incorporate this in Appendix C [now E.2], within the next few days.

---

> ### Author Response · Authors · 2024-11-22
> **Response to g3YX (part II)**
>
> > *“In addition, there are no ablation studies (e.g., one such study could vary the simulation budget or use an adaptive solver for better-than-uniform sampling, another study can consider extremely long time horizons T)”*
>
> We do perform a number of ablation studies: Specifically, we already ablate on different relevant diffusion solvers and composition rules on a larger up of transitions (Fig. 7, [edit now Fig. 10]). Including additional adaptive solvers is out of the scope of this study as we do not aim or claim to improve standard diffusion samplers. We additionally ablate on proposals (Fig. 8 [edit now Fig. 11]), and partially factorized approaches (Fig. 6 [edit now Fig. 9]).
>
> > *“It may be helpful to clarify some points in Section 2.2 regarding related work on amortized inference. … . Additionally, [6, 7] don't seem to introduce, discuss, or validate amortized methods…”*
>
> We will incorporate the suggestions  [2,3,4,5]  into our manuscript and thereby expand our literature review to also include amortized inference in a likelihood-based setting.  In contrast, our work is situated within the context of SBI, where we consider "likelihood-free" or "black-box" scenarios.
>
> ### On Questions
>
> > *"Why does sW seem to, counterintuitively, increase for increasing the number of transitions (Figure 2)?"*
>
> The x-axis corresponds to the number of transitions used to generate the test observations on a fixed training budget (the length of the test time series). The training budget is fixed for all methods to achieve an overall 10k (or 100k) transition simulation. It is expected that the sW distance on all methods increases to some degree. For factorized methods, some local errors might accumulate. For NPE, the embedding network might not be able to generalize to larger sequences than observed in training (e.g., to 100).
>
> >  *"Will the method be robust to missing or corrupted data?"*
>
> Thank you for highlighting this important point. Our method can indeed handle missing or corrupted data. It relies on the Markov properties of time-series simulators at observation points, which are preserved even when data points are missing. One approach to address this is by amortizing over the time step size, $\Delta t$, allowing the method to "skip" missing observations. This will be discussed in Appendix A.

---

> > ### Comment · Area_Chair_oXvM · 2024-11-26
> >
> > Dear reviewer,
> >
> > Please make to sure to read, at least acknowledge, and possibly further discuss the authors' responses to your comments. Update or maintain your score as you see fit.
> >
> > The AC.

---

> > ### Comment · Reviewer_g3YX · 2024-12-02
> >
> > I have read the other reviews and acknowledged the authors' replies. I also appreciate the additional calibration metrics, the increased test budget of 100 simulations (that related papers also used 10 - 25 test simulations seems a weak argument and orthogonal to the practice in general DL research benchmarking methods on tens of thousands of held out instances), and the more moderate formulation. However, I still maintain that the work is rather incremental, while my concerns regarding performance and calibration on longer time horizons than the unrealistic $T = 100$ remain. Accordingly, I have increased my score from 3 to 5.

---

> > > ### Author Response · Authors · 2024-12-02
> > >
> > > We appreciate the reviewer’s response and the thoughtful reevaluation of our work, and we acknowledge the reviewer's concern. We would like to clarify that our study and similar works often limit the evaluation budget for practical reasons. In typical deep learning research, most of the computing time spent training the network and evaluation is cheap (typically evaluating the loss on test samples). In this line of this research, this is often reversed, with the majority of compute time spent on evaluation rather than training (especially when e.g. dealing with NLE, NRE baselines that involve statistical sampling techniques like MCMC).

---

### Official Review · Reviewer_bzn4 · 2024-11-03

**Soundness:** 2
**Presentation:** 3
**Contribution:** 2
**Rating:** 6
**Confidence:** 3

**Summary:**

This paper proposes a simulation-based inference (SBI) method for state-space models where the transition dynamics is Markovian. The core idea is to simulate many single-state transitions and then aggregate them instead of simulating entire trajectories of time-series, so as to reduce the total number of simulator calls.

**Strengths:**

The paper addresses the relevant problem of computational complexity in SBI which arises when the simulator is costly to sample from.

The experimental evaluation seems thorough, and the results are significant for FNSE.

**Weaknesses:**

My main concern is related to the novelty of the paper. The main technical contribution lies in deriving the factorized NSE method, which as the authors mention, is an extension of the setting of the Geffner et al. (2023) paper to the Markov setting. An in-depth analysis/discussion around selecting the proposal $\tilde{p}(\mathbf x^t)$ and providing recommendations on its choice would have strengthened the paper significantly and added to the technical contribution. The paragraph discussing the proposal is kind of confusing. The authors say that "there is lots of flexibility in the choice of the proposal...", but also that "...designing a good proposal can be challenging". I appreciate the experiments evaluating the sensitivity of the results to the choice of proposal, but it is not clear how the proposals are chosen in the first place.

The clarity of writing can be improved by providing examples, especially when it comes to motivating the setting, the core idea, and the assumptions. For instance, the authors say "...given sufficiently many single-step transitions, it should be possible to infer the global target...". A discussion around the cases in which this does (and does not) hold would help the reader understand the scope of this work. Another example is the requirement that "...the proposed state must be independent of the parameters involved in the current state transition...", where it is not clear how restrictive this requirement is, what kind of cases does it cover, etc. Similarly, some examples of expensive real-world scientific simulators in para 3 of the intro would be nice to have.

On the topic of clarity, I found Section 3.2.2 difficult to follow. In particular, it is not clear when the authors are talking about background of previous work (Geffner 2023 and Linhart 2024), and which part is their contribution. Perhaps some of the discussion regarding the implementation details can be moved elsewhere, as they are not easy to follow without reading those two works anyway.

The paper is presented as a sample-efficient method for Markovian time-series simulators. However, the experiments measure the performance as a function of the number of transitions. To conclude that the proposed method is "simulation-efficient", I would have expected performance to be plotted as a function the number of calls to the simulator (or computational cost) for a fixed number of transitions.

Would be nice to have a small discussion on other sample-efficient SBI methods in the related works (e.g. based on sequential sampling, Bayesian optimization, etc.).

Minor comment: Typo in line 125-126 (the word "approaches")

**Questions:**

* Is there a reason why comparison with NSE is not done?
* Can FNSE be also made sequential in a similar manner as other simulation-efficient SBI methods such as SNPE/SNLE/SNRE?
* Which score composition method is used in the experiments when implementing FNSE? (I might have missed if this is mentioned in the appendix)

---

> ### Author Response · Authors · 2024-11-22
> **Response to reviewer bzn4**
>
> We thank the reviewer for the detailed review and valuable feedback on our manuscript. We are happy that the reviewer recognizes the relevance of addressing computational complexity in simulation-based inference (SBI) for costly simulators and appreciates the “thoroughness” and “significance” of our experimental results for FNSE. We acknowledge the concerns and suggestions, and we aim to improve our paper accordingly. Below, we address each of your points in detail.
> ### On Weaknesses
> > *"An in-depth analysis/discussion around selecting the proposal and providing recommendations on its choice would have strengthened the paper"*
>
> We thank the reviewer for their valuable feedback. We agree that the current manuscript does not sufficiently discuss the construction of the proposal. The proposal is task-specific. It can, e.g., be chosen by simple domain knowledge (or observations available beforehand). Within the main manuscript, we use rather “naive” proposals, e.g., given a SIR model, we know that dynamics will be bounded between population size and zero, so we use a Uniform before that interval. This might not be applicable; if such knowledge is unavailable, then we might have to use the simulator within construction. This would require an upfront design cost but leads to better proposals. In response to your comments, we have provided a more general guideline to construct the proposal (see [here](https://filebin.net/mr0aicst9v8p44hb), Sec. 3). And investigated this approach in the Lotka Volterra model, which demonstrated that the upfront cost of designing a proposal via simulations could improve performance even if this cost is subtracted from training budget. This will also be included in the revised manuscript, and we aim to improve our presentation within Sec. 3.
>
> > *"The clarity of writing can be improved"*
>
> Thank you for your insightful suggestion. We fully agree that including examples will help readers better understand our paper. In response, we will add examples to Section 1 to clarify the scope and assumptions of the paper and aim to incorporate your suggestions.
>
> > *"I found Section 3.2.2 difficult to follow."*
>
> We acknowledge that the current manuscript can be difficult to follow without referencing prior works. In response to your suggestion, we will address the following points in the revised paper:
> - Ensure the explanations and notations are self-contained.
> - Move certain technical details, which are not essential for understanding the main content, to the Appendix.
>
> > *“Would be nice to have a small discussion on other sample-efficient SBI methods in the related works”*
>
> We agree that this should be mentioned and will include it in the revised version.
>
> > *“ I would have expected performance to be plotted as a function the number of calls to the simulator”*
>
> We perform evaluations on different simulation budgets  (number of total training transition steps), and within all panels, the budget across methods is the same, allowing discussion about simulation efficiency. We decided to plot it in terms of transition because it is important that the performance translates to a variable-sized time series. Yet, we agree that plotting by simulation budget is more common in related literature. We thus will provide such a figure in the appendix, in addition to our current approach (See [here](https://filebin.net/mr0aicst9v8p44hb), Sec. 4, Fig.4).
>
> ### On Questions
>
> > *“Is there a reason why comparison with NSE is not done?”*
>
> No, we included it as an additional baseline within the Appendix (see [here](https://filebin.net/mr0aicst9v8p44hb), Sec 2). This performed similarly to NPE, as expected, with small improvements in some tasks.
>
> >*”Can FNSE be also made sequential in a similar manner as other simulation-efficient SBI methods such as SNPE/SNLE/SNRE?”*
>
> Yes. Sequential methods in SBI usually target only a posterior for a specific observation $x_o^{0:T}$, whereas we here aim for an amortized approximation. Notably, a rather good proposal for a specific observation (or a set of observations) is to propose $x_o^t$ for random $t$ (optionally with noise). The proposal $p(x_t)$ can also be updated over time in “rounds” without requiring corrections. In addition, one can, for example, use truncated sequential methods for FNSE, which was the only method found effective in the context of score matching (Sharrok et al. 2024). This method does truncate the prior based on the “posterior support” (usually some 99% highest density regions), which allows training without additional correction. Yet, within our framework, this truncation must be based on the “joint” support of the “local” posteriors instead of the global posterior support (as this is our training target). We leave this discussion to future work.
>
> >*”Which score composition method is used in the experiments when implementing FNSE?“*
>
> We used the GAUSS method if not explicitly stated otherwise.

---

> > ### Comment · Reviewer_bzn4 · 2024-11-25
> >
> > Thanks for your response and the additional experiments.
> >
> > * **Regarding technical novelty:** I understand your argument in the global response that the "the effect of this observation is not incremental but rather substantial and increases the applicability of score-based SBI". However, my comment was not about the effect or impact of your insight on the field of SBI, but rather the technical novelty of the proposed method, which I still think is incremental (I wouldn't bring this point up if this was a TMLR submission). For instance, and I am making this example up, imagine if someone empirically shows that a linear model (or something super simple) works as well as all the fancy SBI methods on many problems, such a paper would have big impact on the community. But it would be very difficult to publish such a paper in ML conferences.
> >
> >     Nevertheless, the new discussion on constructing the proposal distribution helps ameliorate this issue a bit, so that's good and improves the manuscript. I would expect this variant of your method with the proposal selected in this new manner be also included in Table 1 for instance, to show the difference between FNSE with and without domain knowledge-based proposal.
> >
> > * **Regarding clarity and other writing related comments:** It is difficult to understand what these responses mean when I cannot read what the updated paper looks like to see if the clarity has improved or not. For instance, you say that you will "Ensure the explanations and notations are self-contained", but how?, or "Move certain technical details...to appendix", but which details?, or which related works are you going to include? It is difficult to modify the review based on such responses, especially when ICLR allows updating the paper.
> >
> > * **Regarding performance as a function of number of calls to the simulator:** I see your point, and thanks for adding that experiment in Figure 4. My comment was that if you want to claim that your method is sample-efficient, then you should also plot performance as a function of simulator calls.
> >
> > Overall, I am happy that the authors discuss and provide recommendations on selecting the proposal (which will hopefully be added to all the experiments), and the additional experiments and comparisons they did. If I could just see some signal that the paper's writing has improved, I would be happy to raise my score to 6.

---

> > > ### Comment · Reviewer_bzn4 · 2024-11-27
> > > **Comments after paper revision**
> > >
> > > Dear Authors,
> > >
> > > Thank you for providing the revised manuscript with the changes highlighted. The methodology section reads much better now, and the related works help put this work in context. I am adding some references below for other compute-efficient methods in SBI that are relevant for the related work section:
> > >
> > >     Multi-fidelity approach: Warne, D. J., Prescott, T. P., Baker, R. E., and Simpson, M. J. (2022). Multifidelity multilevel Monte Carlo to accelerate approximate Bayesian parameter inference for partially observed stochastic processes. Journal of Computational Physics, 469:111543.
> > >
> > >     Using quasi-Monte Carlo sampling: Niu, Z., Meier, J., and Briol, F.-X. (2023). Discrepancy-based inference for intractable generative models using quasi-Monte Carlo. Electronic Journal of Statistics, 17(1):1411–1456.
> > >
> > >     Using cost function of the simulator: Bharti, A., Huang, D., Kaski, S., and Briol, F.-X. (2024). Cost-aware simulation-based inference. arxiv:2410.07930.
> > >
> > >     Method based on early stopping: Prangle, D. (2016). Lazy ABC. Statistics and Computing, 26:171–185.
> > >
> > > In light of the changes to the manuscript, and with the hope that the variant of the proposed method with the proposal constructed as per Section 3.4 will be added to the results, I am increasing my score. I hope our exchange was helpful for the authors.

---

> > > > ### Author Response · Authors · 2024-11-27
> > > >
> > > > We sincerely thank the reviewer for reading our revision and adjusting your score. We are happy that the revisions addressed the concerns on the clarity of writing. Thank you also for introducing the additional related papers, which we included in the related work in the most recent revision.
> > > >
> > > > Regarding the numerical demonstrations of our proposed construction, we have already included them in Appendix C. In addition to the experiment on the Lotka-Volterra task, we also already included the proposal construction for the other tasks in the previous revision (as shown in Tables 1 and 2 of Appendix C, and discussed it in Sec.4.2,4.3). These confirmed that the proposed construction achieves performance that is comparable to or better than the default proposal.
> > > >
> > > > Thank you once again for your thoughtful and detailed engagement with our work, which has significantly improved our submission.

---

### Author Response · Authors · 2024-11-22
**General Response**

We sincerely appreciate the reviewers' efforts in reviewing our manuscript. Our paper introduces a simulation-based inference method for high-dimensional time-series simulators, addressing a "timely (Z2NW)," "challenging (g3YX, MmYK)," and "relevant (bjS2, g3YX, bzn4)" problem. Reviewers found our approach to be "conceptually simple (Ck63)",  "intuitive (bjS2)," "attractive (Ck63)", and "original (g3YX)." They commended our evaluation as "thorough (bzn4)," "significant (bzn4)," "well-motivated (Z2NW)," and "convincing (MmYK)," and "to considering a set of familiar but interesting models (MmYK)."  Finally, the presentation was highlighted as "well-written (Z2NW, bjS2)" and "easy to follow (g3YX, bjS2)," with Z2NW calling it "exceptionally well-written."

Two reviewers (g3YX, bzn4) regarded the contribution of the manuscript as incremental, as they argued that the step from composing scores across independent observations (as in previous work: Geffner et al. 2023, Linhart et al. 2024) to a Markov model (as in our work) was not a substantial contribution. We disagree with this characterization: First, we are very happy to admit and transparently describe that the core idea of the paper is based on the observation that scores can not only be combined across single observations but to pairs of observations and can thus be chained together to perform inference on time-series, while only simulating shorter segments (or, in the extreme case, pairwise transitions). However, while this insight might be simple (in hindsight-- it is novel!), the effect of this observation is not incremental but rather substantial and increases the applicability of score-based SBI. In particular, there are many simulators that can be written as Markov models: most mechanistic simulators of time series (e.g., most discretizations of SDEs, or recurrent models) can be written as Markov models, and previous SBI methods have been quite inefficient in such models. Our simple (in hindsight) insight leads to a substantial increase in simulation-efficiency and applicability of the approach-- we do not agree this is a “very incremental” contribution! Second, and as the paper shows, there are still many technical questions that need to be addressed to make this approach work and validate it, which we address in the paper both empirically and theoretically.

There were also some concerns about the robustness of the evaluation (g3YX) and the lack of baselines (bjS2). In response, we have evaluated and extended our empirical evaluation, which ([here](https://filebin.net/mr0aicst9v8p44hb), Sec. 1,2,4). This includes an additional calibration analysis, which aligns with our current evaluations (on Lotka Volterra and SIR) and up to seven additional baselines (NSE and various variants of NPE, NLE using sufficient neural summary statistics). While some of the additional baselines slightly improved upon our original one, overall, they performed similarly and were distinct from the factorized approaches. Reviewers Z2NW, bzn4, and MmYK pointed out that the selection of proposals is important in practice and that this discussion falls short in the current manuscript. We agree that this was not sufficiently explained in the current submission. In response to the reviewers' comments,  we have provided the two practical guidelines for constructing proposals (see [here](https://filebin.net/mr0aicst9v8p44hb), Sec. 3), which will also be included in the revised manuscript.

We hope that this addresses the reviewer's major concerns, and we plan to upload the revised manuscript next week, which also incorporates suggestions regarding the clarity (bzn4) and presentation (Ck63, MmYK) of the writing.

---

### Comment · Area_Chair_oXvM · 2024-11-26

Dear all,

The deadline for the authors-reviewers phase is approaching (December 2).

@For reviewers, please read, acknowledge and possibly further discuss the authors' responses to your comments. While decisions do not need to be made at this stage, please make sure to reevaluate your score in light of the authors' responses and of the discussion.

- You can increase your score if you feel that the authors have addressed your concerns and the paper is now stronger.
- You can decrease your score if you have new concerns that have not been addressed by the authors.
- You can keep your score if you feel that the authors have not addressed your concerns or that remaining concerns are critical.

Importantly, you are not expected to update your score. Nevertheless, to reach fair and informed decisions, you should make sure that your score reflects the quality of the paper as you see it now. Your review (either positive or negative) should be based on factual arguments rather than opinions. In particular, if the authors have successfully answered most of your initial concerns, your score should reflect this, as it otherwise means that your initial score was not entirely grounded by the arguments you provided in your review. Ponder whether the paper makes valuable scientific contributions from which the ICLR community could benefit, over subjective preferences or unreasonable expectations.

@For authors, please respond to remaining concerns and questions raised by the reviewers. Make sure to provide short and clear answers. If needed, you can also update the PDF of the paper to reflect changes in the text. Please note however that reviewers are not expected to re-review the paper, so your response should ideally be self-contained.

The AC.

---

### Author Response · Authors · 2024-11-26
**Paper Revision**

We appreciate the reviewers' valuable comments and suggestions, which helped us to improve the manuscript. Based on the feedback, we modified our submission, as previously discussed. Major revised points are as follows:

- Extended evaluation: More baselines, calibration analysis, and results on varying training budgets and an extended evaluation of the proposal (as shared previously). These new results are now discussed in the manuscript and added to the Appendix.
- Method section (proposal): We have restructured this section and expanded the discussion of the proposal in the main manuscript. In addition, we added a more detailed and self-contained discussion of both the proposal and score composition methods in the Appendix.
- Related work: As suggested by reviewers, we now also discuss sequential training variants, amortized inference with explicit likelihoods, and SBI with summary statistics.

In the current revised version, we highlighted major revisions or new content in blue. We would ask all the reviewers to reevaluate our work based on the revised manuscript and ideally respond to our rebuttal comments (if not already done).

If there are further suggestions/questions about our submission, we are happy to address them in the remaining days and plan to upload the final revision until then.

Best regards,

the authors

---

### Meta-Review · Area_Chair_oXvM · 2024-12-20

**Metareview:**

The reviewers recommend acceptance (6-5-8-6-6-6). The paper presents a simulation-based inference approach for Markovian state-space models, leveraging the structure of the forward model to reduce the computational cost of the inference. The approach is well-motivated and the results are convincing. The author-reviewer discussion has been constructive and has led to a number of clarifications and improvements, with a better discussion of the related work and the addition of new results. The main concern raised by the reviewers is the incremental contribution of the paper, as the key idea appears to be a straightforward (but novel) extension of previous works. Nevertheless, the reviewers agree that the paper is well-executed and that the results are convincing. For these reasons, I recommend acceptance. I encourage the authors to address the remaining concerns and to implement the modifications discussed with the reviewers in the final version of the paper.

**Additional Comments On Reviewer Discussion:**

The author-reviewer discussion has been constructive and has led to a number of clarifications and improvements, with a better discussion of the related work and the addition of new results.

---

### Decision · Program_Chairs · 2025-01-22

Accept (Poster)